# Positive Unlabeled Learning Selected Not At Random (PULSNAR): class proportion estimation when the SCAR assumption does not hold

## Abstract

Positive and Unlabeled (PU) learning is a type of semi-supervised binary classification where the machine learning algorithm differentiates between a set of positive instances (labeled) and a set of both positive and negative instances (unlabeled). PU learning has broad applications in settings where confirmed negatives are unavailable or difficult to obtain, and there is value in discovering positives among the unlabeled (e.g., viable drugs among untested compounds). Most PU learning algorithms make the *selected completely at random* (SCAR) assumption, namely that positives are selected independently of their features. However, in many real-world applications, such as healthcare, positives are not SCAR (e.g., severe cases are more likely to be diagnosed), leading to a poor estimate of the proportion, $\alpha$, of positives among unlabeled examples and poor model calibration, resulting in an uncertain decision threshold for selecting positives. PU learning algorithms can estimate $\alpha$ or the probability of an individual unlabeled instance being positive or both. We propose two PU learning algorithms to estimate $\alpha$, calculate calibrated probabilities for PU instances, and improve classification metrics: i) PULSCAR (positive unlabeled learning selected completely at random), and ii) PULSNAR (positive unlabeled learning selected not at random). PULSNAR uses a divide-and-conquer approach that creates and solves several SCAR-like sub-problems using PULSCAR. In our experiments, PULSNAR outperformed state-of-the-art approaches on both synthetic and real-world benchmark datasets.

## 1 Introduction

In a standard binary supervised classification problem, the classifier (e.g., decision trees, support vector machines, etc.) is given training instances $\mathcal{X}$ with features $x$ and their labels $y = 0$ (negative) or $y = 1$ (positive). The classifier learns a model $f : \mathcal{X} \to 0, 1$, which classifies an unlabeled instance as positive or negative based on $x$. It is often challenging, expensive, and even impossible to annotate large datasets in real-world applications Jaskie et al. (2019), and frequently only positive instances are labeled. Unlabeled instances with their features can be classified via positive and unlabeled (PU) learning Jaskie et al. (2019); Elkan & Noto (2008). Some of the PU learning literature focuses on improving classification metrics, and others focus on the problem of estimating the fraction, $\alpha$, of positives among the unlabeled instances. Although this work focuses on the latter, calibration and enhancing classification performance are also addressed.

PU learning problems abound in many domains Jaskie & Spanias (2019). For instance, in electronic healthcare records, the lack of a diagnosis code doesn't confirm a patient's negative disease status, as negatives are not routinely recorded, making traditional supervised learning impractical. Much medical literature is dedicated to estimating disease incidence and prevalence but contends with incomplete medical assessment and recording. The potential to assess disease incidence without costly in-person assessment or chart reviews could have substantial public health benefits. In market research, one typically has a modest set of positives, say of customers or buyers of a product, has a set of attributes over both the positives and a large population of unlabeled people of size $N$, and wishes to establish the size of the addressable market, $\alpha N$.

The majority of PU learning algorithms use the *selected completely at random* (SCAR) assumption, which states that the labeled positive examples are randomly selected from the universe of positives. That is, the labeling probability of any positive instance is constant Elkan & Noto (2008). This assumption may fail in real-world applications. For example, in email spam detection, positive instances labeled from an earlier time period could differ from later spam due to adaptive adversaries.

Although some PU learning algorithms have shown promising performance on different machine learning (ML) benchmark SCAR datasets, the development of PU learning algorithms to estimate the extent of undercoding in large and highly imbalanced *selected not at random* (SNAR) real-world data remains an active research area. Class imbalance in a PU setting generally means the number of unlabeled instances is large compared to the labeled positive examples. Also, current PU learning approaches have rarely explored how to calculate well-calibrated probabilities for PU examples in SCAR and SNAR settings. In addition, few PU algorithms have been assessed when $\alpha$ is small ($\leq 5\%$), where performance is expected to suffer.

In this paper, we propose a PU learning approach to estimate $\alpha$ when positives are SCAR or SNAR, and evaluating its performance in simulated and real data. We assess the performance with class imbalance in both modest and large datasets and over a rigorous $\alpha$ range. Our contributions are summarized as follows:

1. We propose PULSCAR, a PU learning algorithm for estimating $\alpha$ when the SCAR assumption holds. It uses kernel density estimates of the positive and unlabeled distributions of ML probabilities to estimate $\alpha$. The algorithm employs the beta distribution to estimate density and introduces an objective function whose derivative maximum provides a rapid, robust estimate of $\alpha$.

2. We propose PULSNAR, a PU learning algorithm for estimating $\alpha$ when the positives are SNAR, that uses a novel clustering approach to divide the positives into several subsets that can have separate $\alpha$ estimates versus the unlabeled. These sub-problems are more SCAR-like and are solved with PULSCAR.

3. We propose methods to calibrate the probabilities of PU examples to their true (unknown) labels and improve the classification performance in SCAR and SNAR settings.

## 2 Related work

Early PU learning methods Yu et al. (2004); Yu (2005); Wang et al. (2006) generally followed a two-step heuristic: i) identify strong negative examples from the unlabeled set, and then ii) apply an ML algorithm to given positive and identified negative examples. In contrast, Fung et al. (2005) extracted high-quality positive and negative examples from the unlabeled set and then applied classifiers to those data. Some recent work iteratively identifies better negatives Luo et al. (2021), or combines negative-unlabeled learning with unlabeled-unlabeled learning Hammoudeh & Lowd (2020).

The following studies in PU learning focused on estimating the proportion of positives among the unlabeled examples with/without the PU classifier. These studies predominantly centered around the SCAR assumption. Elkan & Noto (2008) introduced the SCAR assumption and proposed a PU method to estimate the mixture proportion under the SCAR assumption. By partially matching the class-conditional density of the positive class to the input density under Pearson divergence minimization, Du Plessis & Sugiyama (2014) estimated the mixture coefficient. Jain et al. (2016) proposed a nonparametric class prior estimation technique, AlphaMax, using two-component mixture models. The kernel embedding approaches KM1 and KM2 Ramaswamy et al. (2016) showed that the algorithm for mixture proportion estimation converges to the true prior under certain assumptions. Estimating the class prior through decision tree induction (TICE) Bekker & Davis (2018) provides a lower bound for label frequency under the SCAR assumption. Using the SCAR assumption, DEDPUL Ivanov (2020) estimates $\alpha$ by applying a compute-intensive EM-algorithm to probability densities; the method also returns uncalibrated probabilities.

The following studies employed different approaches to learn a classifier from PU data. Lee & Liu (2003) converts PU data learning into a noisy learning problem by designating all unlabeled instances as negatives. They employ a linear function to learn from these noisy examples using weighted logistic regression. Confident learning (CL) Northcutt et al. (2021) combines the principle of pruning noisy data, probabilistic thresholds

to estimate noise, and sample ranking. Multi-Positive and Unlabeled Learning Xu et al. (2017) extends PU learning to multi-class labels. Oversampling the minority class Chawla et al. (2002); Yan et al. (2019) or undersampling the majority class are not well-suited approaches for PU data due to contamination in the unlabeled set; Su et al. (2021) uses a re-weighting strategy for imbalanced PU learning.

Recent studies have focused on labeling/selection bias to address the SCAR assumption not holding. Bekker et al. (2019); Gerych et al. (2022) used propensity scores to address labeling bias and improve classification. Using the propensity score, based on a subset of features, as the labeling probability for positive examples, Bekker et al. (2019) reduced the Selected At Random (SAR) problem into the SCAR problem to learn a classification model in the PU setting. The "Labeling Bias Estimation" approach was proposed by Gong et al. (2021) to label the data by establishing the relationship among the feature variables, ground-truth labels, and labeling conditions.

## 3 Problem Formulation and Algorithms

In this section, we explain: i) the SCAR and SNAR assumptions, ii) our PULSCAR algorithm for SCAR data and PULSNAR algorithm for SNAR data, iii) bandwidth estimation techniques, and iv) method to find the number of clusters in the labeled positive set. Our method to calibrate probabilities and enhance classification performance using PULSCAR/PULSNAR is in Appendix C and D, respectively.

### 3.1 SCAR assumption and SNAR assumption

In PU learning, a positive or unlabeled example can be represented as a triplet $(x, y, s)$ where "$x$" is a vector of attributes, "$y$" the actual class, and "$s$" a binary variable representing whether or not the example is labeled. If an example is labeled ($s = 1$), it belongs to the positive class ($y = 1$) i.e., $p(y = 1|s = 1) = 1$. If unlabeled ($s = 0$), it can belong to either class. Since only positive examples are labeled, $p(s = 1|x, y = 0) = 0$ Elkan & Noto (2008). Under the SCAR assumption, a labeled positive is an independent and identically distributed (i.i.d) example from the positive distribution, i.e., positives are selected independently of their attributes. Therefore, $p(s = 1|x, y = 1) = p(s = 1|y = 1)$ Elkan & Noto (2008).

For a given dataset, $p(s = 1|y = 1)$ is a constant and is the fraction of labeled positives. If $|P|$ is the number of labeled positives, $|U|$ is the number of unlabeled examples, and $\alpha$ is the unknown fraction of positives in the unlabeled set, then

$$p(s = 1) = \frac{|P|}{|P| + |U|} \quad \text{and} \quad p(y = 1) = \frac{|P| + \alpha|U|}{|P| + |U|}$$

$$p(s = 1|y = 1) = \frac{p(y = 1|s = 1)p(s = 1)}{p(y = 1)} = \frac{p(s = 1)}{p(y = 1)} \quad \text{, since } p(y = 1|s = 1) = 1$$

$$= \frac{|P|}{|P| + \alpha|U|}, \text{ which is a constant.} \tag{1}$$

On the contrary, under the SNAR assumption, the probability that a positive example is labeled is not independent of its attributes. Stated formally, the assumption is that $p(s = 1|x, y = 1) \neq p(s = 1|y = 1)$ i.e. $p(s = 1|x, y = 1)$ is not a constant, which can be proved by Bayes' rule (Appendix A).

The SCAR assumption can hold when both labeled and unlabeled positives: a) are not subclass mixtures, sharing similar attributes; b) belong to $k$ subclasses ($1 \ldots k$), with equal subclass proportions in both positive and unlabeled sets. Intra-subclass examples will have similar attributes, whereas the inter-subclass examples may not have similar attributes. E.g., in patients positive for diabetes, type 1 patients will be in one subclass, and type 2 patients will be in another. The SCAR assumption can fail when labeled and unlabeled positives are from $k$ subclasses, and the proportion of those subclasses is different in positive and unlabeled sets. Suppose both positive and unlabeled sets have subclass 1 and subclass 2 positives, and in the positive set, their ratio is 30:70. If the ratio is also 30:70 in the unlabeled set, the SCAR assumption will hold. If it was different, say 80:20, the SCAR assumption would not hold.

### 3.1.1 PU data assumptions

Positive and unlabeled examples in PU data can be either from a single source or two independent sources. In the *single-training-set scenario (one-sample)*, the positive and unlabeled examples are selected from one dataset, and that dataset is an independent and identically distributed (i.i.d.) sample from the actual distribution. In the *case-control scenario (two-sample)*, positive and unlabeled examples are assumed to have originated from two independent datasets, and the unlabeled dataset is an i.i.d. sample from the actual distribution Elkan & Noto (2008); Bekker & Davis (2020).

Our PU algorithms involve running a machine learning model (PU classifier) on a set of combined positive and unlabeled instances, regardless of whether it is a one-sample or two-sample scenario. The goal is to obtain machine learning-predicted probabilities for all instances. Subsequently, we determine $\alpha$, the fraction of positives among the unlabeled examples. However, our algorithms make certain assumptions regarding the PU data, which include the following: 1) Positive examples have correct labels, i.e., no negative example is marked as positive. Only the unlabeled set has a mix of positives and negatives; 2) The unlabeled positive instances have counterparts (examples with similar features) in the labeled positive set.

As mentioned in the study by Kato et al. (2018), the one-sample scenario is a special case of learning from noisy labels where only negative data are contaminated. Thus, in the one-sample scenario, unlabeled data can be regarded as negative-labeled data contaminated by positive data. Our first assumption satisfies the criteria for the one-sample scenario. Additionally, according to Kato et al. (2018) the unlabeled data of the case-control scenario can be made from positive and unlabeled data of the one-sample scenario. Our second assumption allows us to extend the applicability of our approaches to the two-sample scenario. Our approaches will underestimate $\alpha$ in both one-sample and two-sample scenarios if the second assumption does not hold.

### 3.2 Positive and Unlabeled Learning Selected Completely At Random (PULSCAR) Algorithm

Given any ML algorithm, $\mathcal{A}(x)$, that generates $[0\ldots1]$ probabilities for the data based on covariates $x$, let $f_p(x)$, $f_n(x)$, and $f_u(x)$ be probability density functions (PDFs) corresponding to the probability distribution of positives, negatives, and unlabeled respectively. Let $\alpha$ be the unknown proportion of positives in the unlabeled, then

$$f_u(x) = \alpha f_p(x) + (1-\alpha)f_n(x), \quad \text{using the law of total probability}$$
$$\Rightarrow 1 = \frac{\alpha f_p(x)}{f_u(x)} + \frac{(1-\alpha)f_n(x)}{f_u(x)}$$
$$\Rightarrow \frac{\alpha f_p(x)}{f_u(x)} = 1 - \frac{(1-\alpha)f_n(x)}{f_u(x)}$$
$$\Rightarrow 0 \leq \frac{\alpha f_p(x)}{f_u(x)} \leq 1, \text{ since } 0 \leq \alpha \leq 1 \text{ and } f_n(x) \leq f_u(x)$$
$$\Rightarrow 0 \leq \alpha f_p(x) \leq f_u(x) \tag{2}$$

From property 2, a key observation is that $\alpha f_p(x)$ should not exceed $f_u(x)$ anywhere, allowing one to place an upper bound on $\alpha$.

PULSCAR estimates $\alpha$ by finding the value $\alpha$ where the following objective function maximally changes:

$$f(\alpha) = log(|\min(f_u(x) - \alpha f_p(x))| + \epsilon), \text{ where } \epsilon = |\min(f_p(x))| \text{ if } \min(f_p(x)) \neq 0, \text{ else } \epsilon = 10^{-10} \tag{3}$$

Property 2, $\alpha f_p(x) \leq f_u(x)$, guided the reasoning behind the design choice of the objective function. The intuition behind the objective function is that $|\min(f_u(x) - \alpha f_p(x))|$ approaches zero at the point where $\alpha f_p(x)$ equals $f_u(x)$, see Figure 1D. When we take the logarithm of $|\min(f_u(x) - \alpha f_p(x))|$, the resulting value tends toward $-\infty$ as $|\min(f_u(x) - \alpha f_p(x))|$ approaches zero. Consequently, when $|\min(f_u(x) - \alpha f_p(x))|$ is

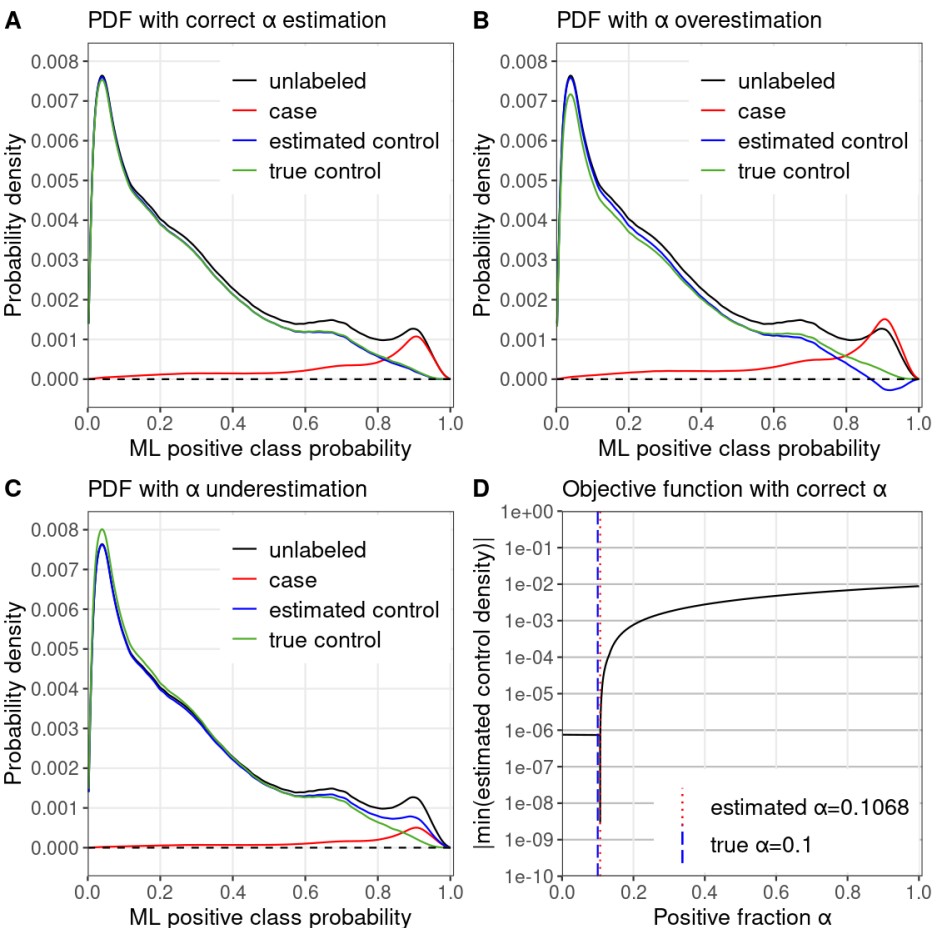

Figure 1: **PULSCAR algorithm visual intuition**. PULSCAR finds the smallest $\alpha$ such that $f_u(x) - \alpha f_p(x)$ is everywhere positive in $[0 \ldots 1]$. A) Kernel density estimates for simulated data with $\alpha = 10\%$ positives in the unlabeled set – estimated negative density (blue) nearly equals the ground truth (green). B) Overweighting the positive density by $\alpha = 15\%$ results in the estimated negative density (blue), $f_u(x) - \alpha f_p(x)$ dropping below zero. C) Underweighting the positive density by $\alpha = 5\%$ results in the estimated negative density (blue) being higher than the ground truth (green). D) Objective function with estimated $\alpha = 10.68\%$ selected where the finite-differences estimate of the slope is largest – very close to ground truth $\alpha = 10\%$.

not zero, there is a maximum change in the value of $\log(|\min(f_u(x) - \alpha f_p(x))|)$ (from $-\infty$ to some value). That is why we locate the point where $\alpha f_p(x)$ equals $f_u(x)$ to place an upper bound on $\alpha$. The reason we use the logarithm is it steeply approaches $-\infty$ as $|\min(f_u(x) - \alpha f_p(x))|$ approaches zero. Adding $\epsilon$ to prevent $log(0)$ and using finite differences to find the max change in slope gives a robust estimator that is resilient to noise. The objective function may not be convex; if multiple points with the same maximal change occur, we take the one closest to zero as the $\alpha$ estimate. This approach eliminates the need for implementing an iterative solver technique, accounting in part for the speed of our algorithm.

We use beta kernel density estimates on ML-predicted class 1 probabilities of positives and unlabeled to estimate $f_p(x)$ and $f_u(x)$. We use a finite difference approximation of the slope of $f(\alpha)$ to find its maximum. The value of $\alpha$ can also be determined visually by plotting the objective function (Figure 1D); the sharp inflection point in the plot represents the value of $\alpha$. Algorithm 1 shows the pseudocode of the PULSCAR algorithm to estimate $\alpha$ using the objective function based on probability densities. Algorithm 2 is a subroutine to compute the beta kernel bandwidth. Full source code for our algorithms is provided as a supplemental document.

**Algorithm 1** PULSCAR Algorithm
___
**Input**: X $(X_p \cup X_u)$, y $(y_p \cup y_u)$, n_bins
**Output**: estimated $\alpha$

 1: predicted_probabilities (p) $\leftarrow \mathcal{A}(X, y)$
 2: p0 $\leftarrow$ p[y == 0]
 3: p1 $\leftarrow$ p[y == 1]
 4: estimation_range $\leftarrow$ [0, 0.0001, 0.0002, ..., 1.0]
 5: bw $\leftarrow$ estimate_bandwidth_pu(p, n_bins)
 6: $D_u \leftarrow$ beta_kernel(p0, bw, n_bins)
 7: $D_p \leftarrow$ beta_kernel(p1, bw, n_bins)
 8: $\epsilon \leftarrow |\min(D_p)|$
 9: **if** $\epsilon = 0$ **then**
10: $\quad \epsilon \leftarrow 10^{-10}$
11: **end if**
12: len $\leftarrow$ length(estimation_range)
13: selected_range $\leftarrow$ estimation_range[2:len]
14: $\alpha \leftarrow$ estimation_range
15: f($\alpha$) $\leftarrow \log(|\min(D_u - \alpha D_p)| + \epsilon)$
16: d $\leftarrow$ f'($\alpha$)
17: i $\leftarrow$ where the value of d changes maximally
18: **return** selected_range[i]

**Algorithm 2** estimate_bandwidth_pu
___
**Input**: predicted_probabilities, n_bins
**Output**: bandwidth

 1: preds $\leftarrow$ predicted_probabilities
 2: bw $\in$ [0.001, 0.5]
 3: $D_{hist} \leftarrow$ histogram(preds, n_bins, density=True)
 4: $D_{beta} \leftarrow$ beta_kernel(preds, bw, nbins)
 5: **return** optimize(MeanSquaredError($D_{hist}$, $D_{beta}$))

### 3.3 Kernel Bandwidth estimation

A beta kernel estimator is used to create a smooth density estimate of both the positive and unlabeled ML probabilities, generating distributions over $[0 \ldots 1]$, free of the problematic boundary biases of kernels (e.g. Gaussian) whose range extends outside that interval, adopting the approach of Chen (1999). Another problem with (faster) Gaussian kernel density implementations is that they often use polynomial approximations that can generate negative values in regions of low support, dramatically distorting $\alpha$ estimates which require non-negative probability distribution estimates. The beta PDF is as follows Virtanen et al. (2020):

$$h(x, a, b) = \frac{\Gamma(a + b) x^{a-1} (1 - x)^{b-1}}{\Gamma(a)\Gamma(b)}, \tag{4}$$

for x $\in$ [0,1], where $\Gamma$ is the gamma function, $a = 1 + \frac{z}{bw}$ and $b = 1 + \frac{1-z}{bw}$, with $z$ the bin location, and $bw$ the bandwidth.

Kernel bandwidth selection can also significantly influence $\alpha$ estimates: too narrow of a bandwidth can result in outliers driving poor estimates, and too wide of a bandwidth prevents distinguishing between distributions. We use a histogram bin count heuristic to generate a histogram density, then optimize the beta distribution bandwidth to best fit that histogram density.

#### 3.3.1 Bin count

Our implementation supports 4 well-known methods to determine the number of histogram bins: square root, Sturges' rule, Rice's rule, Scott's rule, and Freedman–Diaconis (FD) rule Alxneit (2020).

#### 3.3.2 Bandwidth estimation

We compute a histogram density using a bin count heuristic and a beta kernel density estimate at those bin centers using the ML probabilities of both the positive and unlabeled examples. We find the global minimum of the mean squared error (MSE) between the histogram and beta kernel densities using the scipy *differential_evolution()* optimizer Virtanen et al. (2020), solving for the best bandwidth in the range

[0.001...0.5]. That bandwidth is chosen for kernel density estimation in the PULSCAR algorithm. All experiments herein use MSE as the error metric, butthe Jensen-Shannon distance may also be employed.

### 3.4   Positive and Unlabeled Learning Selected Not At Random (PULSNAR) Algorithm

We propose a new PU learning algorithm (PULSNAR) to estimate the $\alpha$ in SNAR data, i.e., labeled positives are not selected completely at random. PULSNAR uses a divide-and-conquer strategy for the SNAR data. It converts a SNAR problem into several sub-problems using an unsupervised learning method (clustering), each of which better approximates the SCAR assumption holding; then applies the PULSCAR algorithm to those sub-problems. The final alpha is computed by summing the alpha returned by the PULSCAR algorithm for each cluster.

$$\alpha = \alpha_1 + \alpha_2 + ... + \alpha_c, \quad c = \text{number of clusters} \tag{5}$$

Figure 2 visualizes the PULSNAR algorithm, and Algorithm 3 provides its pseudocode.

#### 3.4.1   Clustering rationale

Suppose both positive and unlabeled sets contain positives from $k$ subclasses $(1 \ldots k)$. With selection bias (SNAR), the subclass proportions will differ between the sets, and thus the PDF of the labeled positives cannot be scaled by a uniform $\alpha$ to estimate positives among the unlabeled. The smallest subclass would drive an $\alpha$ underestimate with PULSCAR. To address this, we apply clustering to the labeled positives to split them into $c$ clusters. Clustering separates subclasses of positives, and if the assumption that subclass membership drives selection bias holds, PU data comprising examples from one cluster and the unlabeled set will approximate the SCAR assumption. Applying PULSCAR to each cluster of positives versus the unlabeled results in better estimates of the proportions of similar unlabeled positives (Figure 2).

#### 3.4.2   Determining the number of clusters in the positive set

We build an XGBoost Chen & Guestrin (2016) model on all positive and unlabeled examples to determine the important features and their *gain* scores. A *gain* score measures the magnitude of the feature's contribution to the model. We select all labeled positives and then cluster them on those features scaled by their corresponding *gain* score, using scikit_learn's Gaussian mixture model (GMM) method. To establish the number of clusters (n_components), we iterate n_components over $1 \ldots m$ (e.g., $m=25$) and compute the Bayesian information criterion (BIC)Vrieze (2012) for each clustering model. We use max_iter=250, and covariance_type="full". The other parameters are used with their default values. We implemented the "Knee Point Detection in BIC" algorithm, explained in Zhao et al. (2008), to find the number of clusters in the labeled positives.

### 3.5   Calculating calibrated probabilities

An approach to calibrate the ML-predicted probabilities of positive and unlabeled examples in the SCAR and SNAR data is explained in Appendix C.

### 3.6   Improving classification performance

An approach to improving PULSCAR and PULSNAR classification, based on flipping the highest probability $\alpha|U|$ unlabeled examples to 1, is explained in Appendix D.

## 4   Experiments

We evaluated our proposed PU learning algorithms in terms of $\alpha$ estimates, six classification performance metrics, and probability calibration. We used real-world ML benchmark datasets and synthetic data for our experiments. For real-world data, we used Bank Moro et al. (2014) and KDD Cup 2004 particle physics Caruana et al. (2004) datasets as SCAR data and Statlog (Shuttle) UCI ML Repository (2022b) and Firewall

datasets UCI ML Repository (2022a) as SNAR data. The synthetic (SCAR and SNAR) datasets were generated using the scikit-learn function *make_classification()* Pedregosa et al. (2011). We used XGBoost as a binary classifier in our proposed algorithms. To train the classifier on the imbalanced data, we used *scale_pos_weight* parameter of XGBoost to scale the weight of the labeled positive examples by the factor $s = \frac{|U|}{|P|}$. We observed that if the ratio, $s$, of the majority class to the minority class is less than 50, handling class imbalance can be achieved by setting the *scale_pos_weight* parameter of XGBoost to $s$. We also compared our methods with five recently published methods for PU learning: KM1 and KM2 Ramaswamy et al. (2016), TICE Bekker & Davis (2018), DEDPUL Ivanov (2020) and CleanLab Northcutt et al. (2021). KM1, KM2, and TICE algorithms were not scalable and failed to execute on large datasets, so we used smaller synthetic datasets to compare our method with these methods. We compared PULSNAR with only DEDPUL on large synthetic datasets (Appendix E). Also, Ivanov (2020) previously demonstrated that DEDPUL outperformed KM and TICE algorithms on several UCI (University of California Irvine) ML benchmark and synthetic datasets.

---

**Algorithm 3** PULSNAR Algorithm
---
**Input**: X ($X_p \cup X_u$), y ($y_p \cup y_u$), n_bins
**Output**: estimated $\alpha$

1: feature_importance ($v_1...v_k$),
   imp_features ($x_1...x_k$) $\leftarrow \mathcal{A}(X, y)$
2: $x'_1...x'_k \leftarrow x_1 v_1...x_k v_k$
3: $X'_p \leftarrow X_p[x'_1...x'_k]$
4: clusters $s_1...s_c \leftarrow$ GMM($X'_p$)
5: $\alpha \leftarrow 0$
6: **for** c in $s_1...s_c$ **do**
7:     X' $\leftarrow X_p[c] \cup X_u$
8:     y' $\leftarrow y_p[c] \cup y_u$
9:     $\alpha \leftarrow \alpha +$ PULSCAR(X', y', n_bins)
10: **end for**
11: **return** $\alpha$

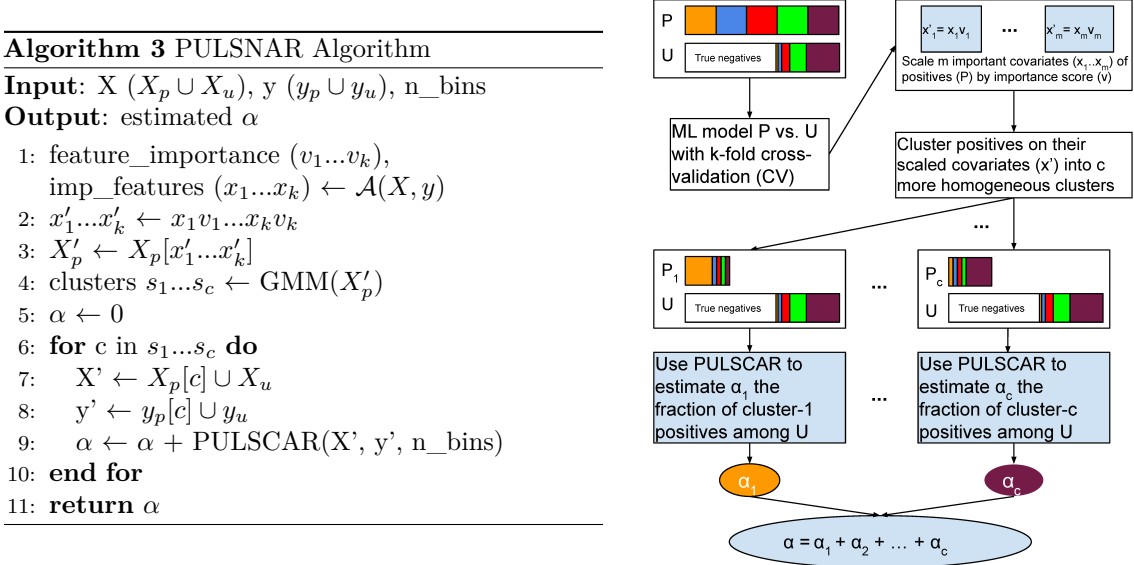

Figure 2: **Schematic of PULSNAR algorithm**. An ML model is trained and tested with 5-fold CV on all positive and unlabeled examples. The model covariates are scaled by their importance value. Positives are divided into c clusters using the scaled important covariates. c ML models are trained and tested with 5-fold CV on the records from a cluster and all unlabeled records. We estimate the proportions ($\alpha_1...\alpha_c$) of each subtype of positives in the unlabeled examples using PULSCAR. The sum of those estimates gives the overall fraction of positives in the unlabeled set. P = positive set, U = Unlabeled set.

## 4.1 Synthetic data

We generated SCAR and SNAR PU datasets with different fractions of positives (1%, 5%, 10%, 20%, 30%, 40%, and 50%) among the unlabeled examples to test the effectiveness of our proposed algorithms. For each fraction, we generated 40 datasets using sklearn's *make_classification()* function with random seeds 0-39. The *class_sep* parameter of the function was used to specify the separability of data classes. Values nearer to 1.0 make the classification task easier; we used class_sep=0.3 to create difficult classification problems.

### 4.1.1 SCAR data

The datasets contained 2,000 positives (class 1) and 6,000 unlabeled (class 0) examples with 50 continuous features. The unlabeled set comprised $k\%$ positive examples with labels flipped to 0 and $(100 - k)\%$ negative examples.

### 4.1.2 SNAR data

We generated datasets with 6 labels (0-5), defining '0' as negative and 1-5 as positive subclasses. These datasets contained 2,000 positives (400 from each positive subclass) and 6,000 unlabeled examples with 50 continuous features. The unlabeled set comprised k% positive examples with labels (1-5) flipped to 0 and (100-k)% negative examples. The unlabeled positives were markedly SNAR, with the 5 subclasses comprising 1/31, 2/31, 4/31, 8/31, and 16/31 of the unlabeled positives. (e.g., in the unlabeled set with 20% positives, negative: 4,800, label 1 positive: 39, label 2 positive: 77, label 3 positive: 155, label 4 positive: 310, label 5 positive: 619).

## 4.2 SCAR ML Benchmark Datasets

### 4.2.1 UCI Bank dataset

The dataset has 45,211 records (class 1: 5,289, class 0: 39,922) with 16 features. This dataset is a good example of data with class imbalance and mixed features. Since the features contain both numerical and categorical values, they were one-hot encoded Hollaar (1982) using the scikit-learn function *OneHotEncoder()* Pedregosa et al. (2011). The encoder derives the categories based on the unique values in each feature, resulting in 9,541 features. The ML classifier was applied to the encoded features.

### 4.2.2 KDD Cup 2004 Particle Physics dataset

The dataset contains two types of particles generated in high-energy collider experiments; 50,000 examples (class 1: 24,861, class 0: 25,139) with 78 numerical attributes. This dataset is a good example of balanced data.

In both datasets, class 1 records were used as positive, and class 0 records were used as unlabeled for the ML model. To add k% positive examples to the unlabeled set, the labels of $m$ randomly selected positive records were flipped from 1 to 0, where $m = \frac{k|U|}{100-k}$.

## 4.3 SNAR ML Benchmark Datasets

### 4.3.1 UCI Statlog (Shuttle) Dataset

The dataset contains 43,500 records (class 1: 34,108, class 2: 37, class 3: 132, class 4: 6,748, class 5: 2,458, class 6: 6, class 7: 11) with 9 numerical attributes. This dataset is an example of data with multiclass and class imbalance. We used class 1 as unlabeled examples and the rest of the records as subclasses of positive examples for the ML model (positive: 9,392, unlabeled: 34,108).

### 4.3.2 UCI Firewall dataset

It is a multiclass dataset containing 65,532 records ('allow': 37,640, 'deny': 14,987, 'drop': 12,851, 'reset-both': 54) with 12 numerical attributes. Class 'allow' was used as unlabeled examples, and the others ('deny', 'drop', 'reset-both') were used as subclasses of positive examples for the ML model (positive: 27,892, unlabeled: 37,640).

In both datasets, the majority of positives are from two classes (*shuttle: class 4, 5; firewall: 'deny', 'drop'*). So, to add $k\%$ positive examples to the unlabeled set, we randomly selected some examples from the minor positive classes and the remaining examples from two major positive classes in equal proportion. Thus, the proportion of positives in the positive set differed from the unlabeled set.

## 4.4 Estimation of fraction of positives among unlabeled examples

We applied the PULSCAR algorithm to both SCAR and SNAR data, and the PULSNAR algorithm only to SNAR data, to estimate $\alpha$.

### 4.4.1 Using the PULSCAR algorithm

To find the 95% confidence interval (CI) on estimation, we ran XGBoost with 5-fold cross-validation (CV) for 40 random instances of each dataset generated (or selected from benchmark data) using 40 random seeds. Each iteration's class 1 predicted probabilities of positives and unlabeled were used to calculate the value of $\alpha$.

### 4.4.2 Using the PULSNAR algorithm

The labeled positives were divided into $c$ clusters to get homogeneous subclasses of labeled positives. The XGBoost ML models were trained and tested with 5-fold CV on data from each cluster and all unlabeled records. For each cluster, $\alpha$ was estimated by applying the PULSCAR method to class 1 predicted probabilities of positives from the cluster and all unlabeled examples. The overall proportion was calculated by summing the estimated $\alpha$ for each cluster. To compute the 95% CI on the estimation, PULSNAR was repeated 40 times on data generated/selected using 40 random seeds.

## 5 Results

### 5.1 Synthetic datasets

Figure 3 shows the $\alpha$ estimated by PU learning algorithms for synthetic datasets. Appendix Figure 6 shows the difference between the mean $\alpha$ estimated by PU learning algorithms and the true fractions for synthetic datasets. TICE overestimated $\alpha$ for all fractions in both SCAR and SNAR datasets. For SCAR datasets, only PULSCAR returned close estimates for all fractions; DEDPUL overestimated for 1%; KM1 and KM2 underestimated for 50%; CleanLab underestimated for larger $\alpha$ (10-50%). For SNAR datasets, only PULSNAR's estimates were close to the true $\alpha$; other algorithms overestimated/underestimated for larger $\alpha$ (30-50%). Figure 15 in Appendix E shows the $\alpha$ estimated by DEDPUL and PULSNAR on large SNAR datasets with different class imbalances. As the class imbalance increased, the performance of DEDPUL dropped, especially for larger fractions. The estimated $\alpha$ by the PULSNAR method was close to the true $\alpha$ for all fractions and sample sizes.

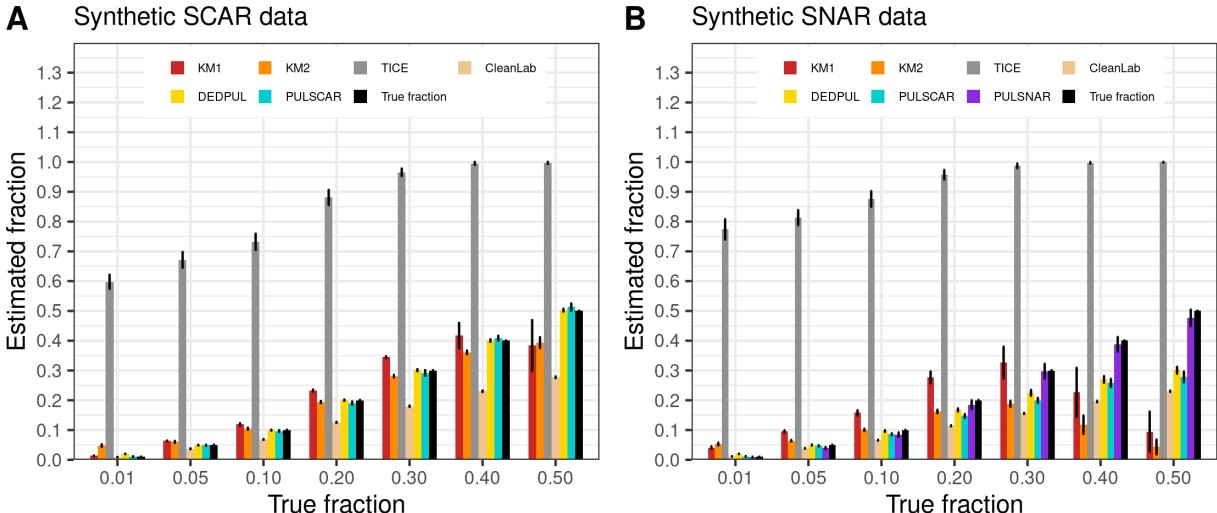

Figure 3: **KM1, KM2, TICE, CleanLab, DEDPUL, PULSCAR, and PULSNAR evaluated on SCAR and SNAR synthetic datasets**. The bar represents the mean value of the estimated $\alpha$, with 95% confidence intervals for estimated $\alpha$.

## 5.2 ML Benchmark datasets

### 5.2.1 SCAR data

Figure 4 shows the $\alpha$ estimated by PU learning algorithms for the KDD Cup 2004 particle physics and UCI bank datasets. For KDD Cup, estimates by PULSCAR and DEDPUL were close to the true answers for all fractions; TICE overestimated for all fractions; CleanLab overestimated for 1-30%. For Bank, only PULSCAR returned correct estimates for all fractions; other algorithms overestimated for all fractions.

### 5.2.2 SNAR data

Figure 5 shows the $\alpha$ estimated by PU learning algorithms for UCI Shuttle and UCI Firewall datasets. For the Shuttle dataset, only PULSNAR's estimates were close to the true fractions; other algorithms either overestimated or underestimated. For the Firewall dataset, TICE overestimated, and CleanLab underestimated for all fractions; PULSNAR's estimates were within $\pm 20\%$ of the true $\alpha$; DEDPUL and PULSCAR underestimated for 40%.

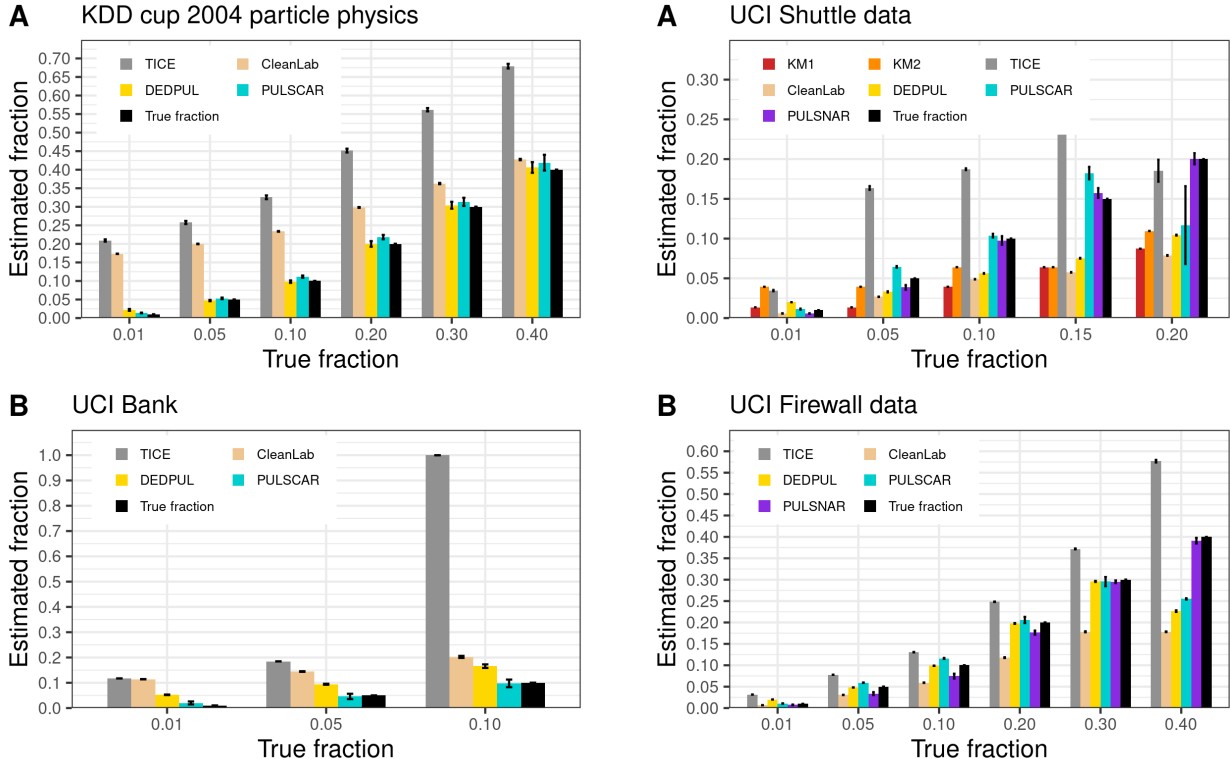

Figure 4: **TICE, CleanLab, DEDPUL, and PULSCAR evaluated on SCAR KDD cup 2004 particle physics and UCI Bank datasets**. The bar represents the mean value of the estimated $\alpha$, with 95% confidence intervals for estimated $\alpha$. KM1 and KM2 failed to execute on both datasets. As TICE was taking several hours to finish one iteration on the bank dataset, the mean $\alpha$ was computed using 5 iterations, and the standard error was set to 0.

Figure 5: **KM1, KM2, TICE, CleanLab, DEDPUL, PULSCAR, and PULSNAR evaluated on SNAR UCI Shuttle and Firewall datasets**. The bar represents the mean value of the estimated $\alpha$, with 95% confidence intervals for estimated $\alpha$. KM1 and KM2 failed to execute on the Firewall dataset. As KM1 and KM2 were taking several hours to finish one iteration on the Shuttle dataset, the mean $\alpha$ was computed using 5 iterations, and the standard error was set to 0.

### 5.3 Probability calibration

Appendix C.2 shows the calibration curves generated using the unblinded labels and isotonically calibrated probabilities of positive and unlabeled examples or only unlabeled examples in the SCAR and SNAR data.

### 5.4 Classification performance metrics

Appendix D.2 shows substantial improvement in 6 classification performance metrics when applying PULSCAR and PULSNAR versus XGBoost alone.

## 6 Discussion and Conclusion

This paper presented novel PU learning algorithms to estimate the proportion of positives among unlabeled examples in both SCAR and SNAR data with/without class imbalance. Preliminary work (not shown) suggests PULSNAR $\alpha$ estimation is robust to overestimating the number of clusters in SNAR data. Our synthetic experiments were run on difficult classification tasks with low separability. For SNAR data, with true $\alpha = 1\%$, when we increased *class_sep* from 0.3 to 0.5 the PULSNAR $\alpha$ estimate improved from 1.6% (Figure 3) to 0.98% (data not shown). Experimentally, we showed that our proposed methods outperformed state-of-art methods on synthetic and real-world SCAR and SNAR datasets. PU learning methods based on the SCAR assumption generally give poor $\alpha$ estimates on SNAR data. We demonstrated that after applying PULSCAR/PULSNAR, classifier performance, including calibration, improved significantly. Since the SCAR assumption often does not hold in real-world data, better $\alpha$ estimates in a SNAR setting open up new horizons in PU Learning.

Our experimentation showed that the KM1, KM2, and TICE algorithms exhibited scalability issues and could not process large datasets with high dimensions. This observation aligns with the findings of Garg et al. (2021), who noted the underperformance of these techniques in high-dimensional scenarios and scalability issues with large datasets. While we evaluated PULSCAR/PULSCAR against these methods using moderately sized datasets, it is plausible that their inherent limitations with data size and high dimension contributed to inaccurate $\alpha$ estimates for some of our test datasets. The CleanLab method is not explicitly designed for PU problems but is primarily developed for noisy label problems. This could be a potential explanation for its poor effectiveness when applied to PU scenarios.

We posit two reasons why PULSCAR outperformed DEDPUL in some experiments: a) The version of PULSCAR presented here uses the Beta distribution, outperforming earlier prototypes that used the Gaussian distribution. This may account for PULSCAR's superiority over DEDPUL, which uses the Gaussian kernel for density estimation. The problems with boundary biases of the Gaussian (as well as the dangers of polynomial approximations to the Gaussian) are described in section 3.3, which are addressed using the Beta. b) In addition, we believe our robust approach to density-based $\alpha$ estimation using Equation 3 may have more robust convergence properties than the EM algorithm used by DEDPUL.

## 7 Limitations

Our approach counts on knowing whether the data are SCAR or SNAR because the knee-point cluster determination approach may produce $> 1$ clusters on SCAR data containing just one type of positive in both positive and unlabeled sets. This will result in PULSNAR overestimating $\alpha$ as two near-identical positive types cannot be distinguished and get counted more than once.

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

# Appendix

## A    Proof: positives are not independent of their attributes under the SNAR Assumption

Under the SNAR assumption, the probability that a positive example is labeled is not independent of its attributes. Stated formally, the assumption is that $p(s = 1|x, y = 1) \neq p(s = 1|y = 1)$ i.e. $p(s = 1|x, y = 1)$ is not a constant.

**Proof:**

$$
\begin{aligned}
p(s = 1|x, y = 1) &= p(y = 1|(s = 1|x))p(s = 1|x) \\
&= p(y = 1|(s = 1|x))\frac{p(x|s = 1)p(s = 1)}{p(x)} \text{ , using Bayes' rule} \\
&= \frac{p(x|s = 1)p(s = 1)}{p(x)} \text{ , since } p(y = 1|(s = 1|x)) = 1 \\
&= \text{a function of } x.
\end{aligned}
$$

## B Errors in $\alpha$ estimation with synthetic SCAR and SNAR data

To further emphasize the magnitude and direction of the $\alpha$ estimate errors in Figure 3, we show the difference between the estimated and true values.

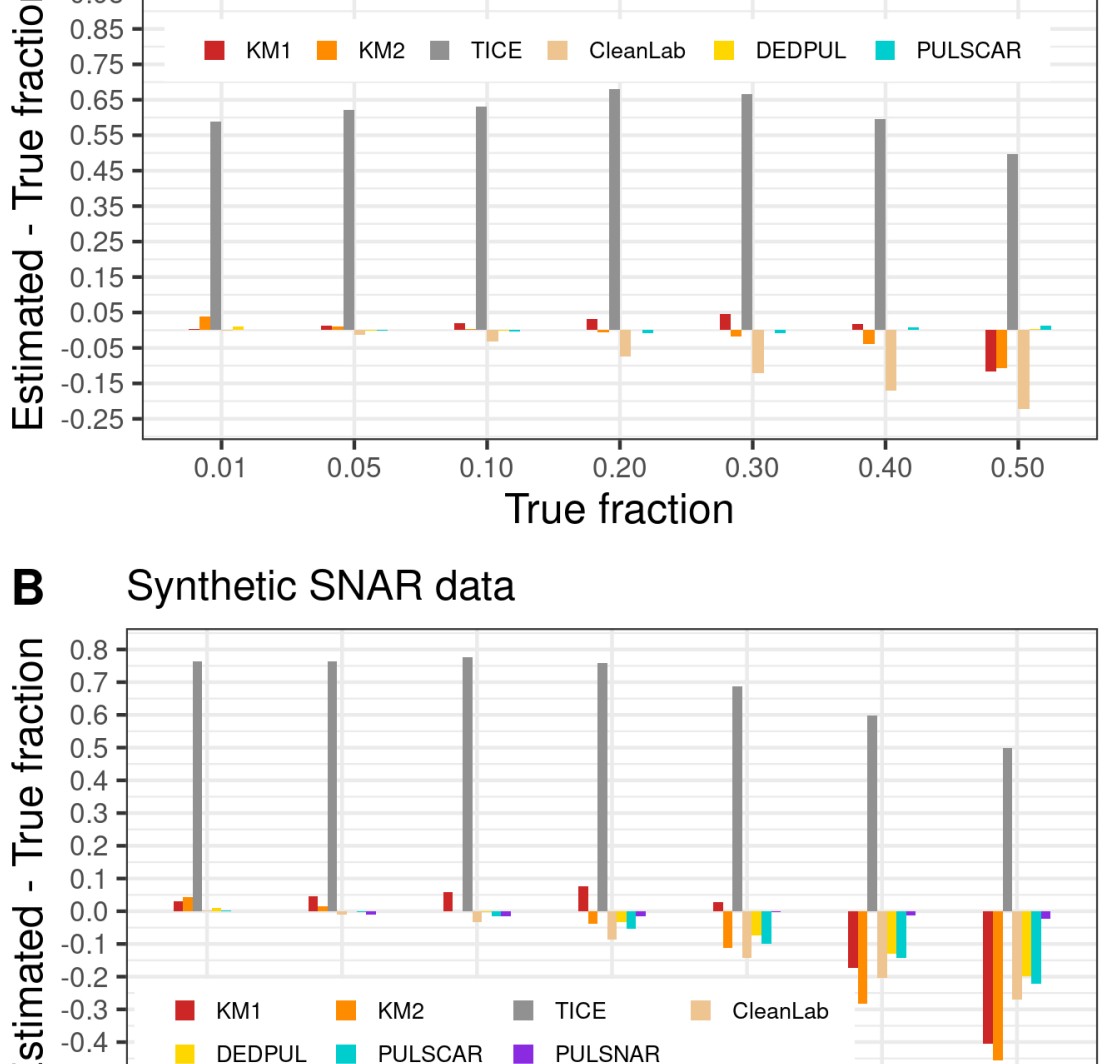

Figure 6: **KM1, KM2, TICE, CleanLab, DEDPUL, PULSCAR, and PULSNAR evaluated on SCAR and SNAR synthetic datasets**. The bar represents the difference between the mean value of the estimated $\alpha$ and the true fraction. Bar above the line y=0 represents overestimation and bar below the line y=0 represents underestimation.

## C  Probability calibration

### C.1  Algorithm for calibrating probabilities

Algorithm 4 shows the complete pseudocode to calibrate the machine learning (ML) model predicted probabilities. Once $\alpha$ is known, we seek to transform the original class 1 probabilities so that their sum is equal to $\alpha|U|$ among the unlabeled or $|P| + \alpha|U|$ among positive and unlabeled, and that they are well-calibrated. Our approach is to probabilistically flip $\alpha|U|$ labels of unlabeled to positive (from 0 to 1) in such a way as to match the PDF of labeled positives across 100 equispaced bins over $[0\ldots1]$, then fit a logistic or isotonic regression model on those labels versus the probabilities to generate the transformed probabilities. To determine the number of unlabeled examples that need to be flipped in each bin, we compute the normalized histogram density, $D\_hist$, for the labeled positives with 100 bins and then multiply $\alpha|U|$ with $D\_hist$.

The unlabeled examples are also divided into 100 bins based on their predicted probabilities. Starting from the bin with the highest probability(p=1), we randomly select $k$ examples and flip their labels from 0 to 1, where $k$ is the number of unlabeled examples that need to be flipped in the bin. If the number of records ($n_1$) that need to be flipped in a bin is more than the number of records ($n_2$) present in the bin, the difference ($n_1 - n_2$) is added to the number of records to be flipped in the next bin, resulting in $\alpha|U|$ flips.

After flipping the labels of $\alpha|U|$ unlabeled examples from 0 to 1, we fit an isotonic or sigmoid regression model on the ML-predicted class 1 probabilities with the updated labels to obtain calibrated probabilities.

The above calibration approach applies to both SCAR and SNAR data. For the SNAR data, the PULSNAR algorithm divides labeled positive examples into $k$ clusters and estimates the $\alpha$ for each cluster. For each cluster, the ML-predicted class 1 probabilities of the examples (positives from the cluster and all unlabeled examples or only unlabeled examples) are calibrated using the estimated $\alpha$ for the cluster. Since, for each cluster, PULSNAR uses all unlabeled examples, each unlabeled example has $k$ ML-predicted/calibrated probabilities. The final ML-predicted/calibrated probability of an unlabeled example is calculated using the following Equation 6:

$$p = 1 - (1 - p_1)(1 - p_2)\ldots(1 - p_k) \tag{6}$$

where $p_k$ is the probability of an unlabeled example from cluster $k$.

### C.2  Experiments and Results

We used synthetic SCAR and SNAR datasets and KDD Cup SCAR dataset to test our calibration algorithm.

**SCAR datasets:** After estimating the $\alpha$ using the PULSCAR algorithm, we applied Algorithm 4 to calibrate the ML-predicted probabilities. To calculate the calibrated probabilities for both positive and unlabeled (PU) examples, we applied isotonic regression to the ML-predicted class 1 probabilities of PU examples with labels of positives and updated labels of unlabeled (of which $\alpha|U|$ were flipped per Algorithm 4). We applied isotonic regression to the unlabeled's predicted probabilities with their updated labels to calculate the calibrated probabilities only for the unlabeled.

**SNAR datasets:** Using the PULSNAR algorithm, the labeled positive examples were divided into $k$ clusters. For each cluster, after estimating the $\alpha$, Algorithm 4 was used to calibrate the ML-predicted probabilities. To calculate the calibrated probabilities for positives from a cluster and all unlabeled examples, we applied isotonic regression to their ML-predicted class 1 probabilities with labels of positives from the cluster and updated labels of unlabeled (of which $\alpha_j|U|$ were flipped for cluster $j = 1\ldots k$, see Algorithm 4). We applied isotonic regression to the unlabeled's predicted probabilities with their updated labels to calculate the calibrated probabilities only for the unlabeled. Thus, each unlabeled example had $k$ calibrated probabilities. We computed the final calibrated probability for each unlabeled example using Formula 6.

Figures 7, 8, 9, 10, 11 and 12 show the calibration curves generated using the unblinded labels and isotonically calibrated (red)/ uncalibrated (blue) probabilities. When both positive and unlabeled examples were used to calculate calibrated probabilities, the calibration curve followed the y=x line (well-calibrated probabilities).

When only unlabeled examples were used, the calibration curve for 1% did not follow the y=x line, presumably due to the ML model being biased toward negatives, given the small $\alpha$. Also, the calibration curves for the SCAR data followed the y=x line more closely than the calibration curves for the SNAR data. It is due to the fact that the final probability of an unlabeled example in the SNAR data is computed using its $k$ probabilities from $k$ clusters. So, a poor probability estimate from even one cluster can influence the final probability of an unlabeled example.

---

**Algorithm 4** calibrate_probabilities

---

**Input**: predicted_probs, labels, n_bins, calibration_method, calibration_data, $\alpha$
**Output**: calibrated_probs

1: p0 $\leftarrow$ predicted_probs[labels == 0]
2: p1 $\leftarrow$ predicted_probs[labels == 1]
3: y0 $\leftarrow$ labels[labels == 0]
4: y1 $\leftarrow$ labels[labels == 1]
5: $D_{hist} \leftarrow$ histogram(p1, n_bins, density=True)
6: unlab_pos_count_in_bin $\leftarrow \alpha\ |p0|\ D_{hist}$
7: p0_bins $\leftarrow$ split unlabeled examples into n_bins using p0
8: **for** k $\leftarrow$ [n_bins ... 1] **do**
9:     $n_1 \leftarrow$ unlab_pos_count_in_bin[k]
10:     $n_2 \leftarrow$ p0_bins[k]
11:     **if** $n_1 > n_2$ **then**
12:         $\hat{y0} \leftarrow$ flip labels (y0) of $n_2$ examples from 0 to 1 in bin k
13:         unlab_pos_count_in_bin[k-1] $\leftarrow$ unlab_pos_count_in_bin[k-1] + $(n_1 - n_2)$
14:     **else**
15:         $\hat{y0} \leftarrow$ flip labels (y0) of random $n_1$ examples from 0 to 1 in bin k
16:     **end if**
17: **end for**
18: **if** calibration_data == 'PU' **then**
19:     p, y $\leftarrow p1 \cup p0, y1 \cup \hat{y0}$
20: **else if** calibration_data == 'U' **then**
21:     p, y $\leftarrow$ p0, $\hat{y0}$
22: **end if**
23: **if** calibration_method is 'sigmoid' **then**
24:     $\hat{p} \leftarrow$ LogisticRegression(p, y)
25: **else if** calibration_method is 'isotonic' **then**
26:     $\hat{p} \leftarrow$ IsotonicRegression(p, y)
27: **end if**
28: **return** $\hat{p}$

---

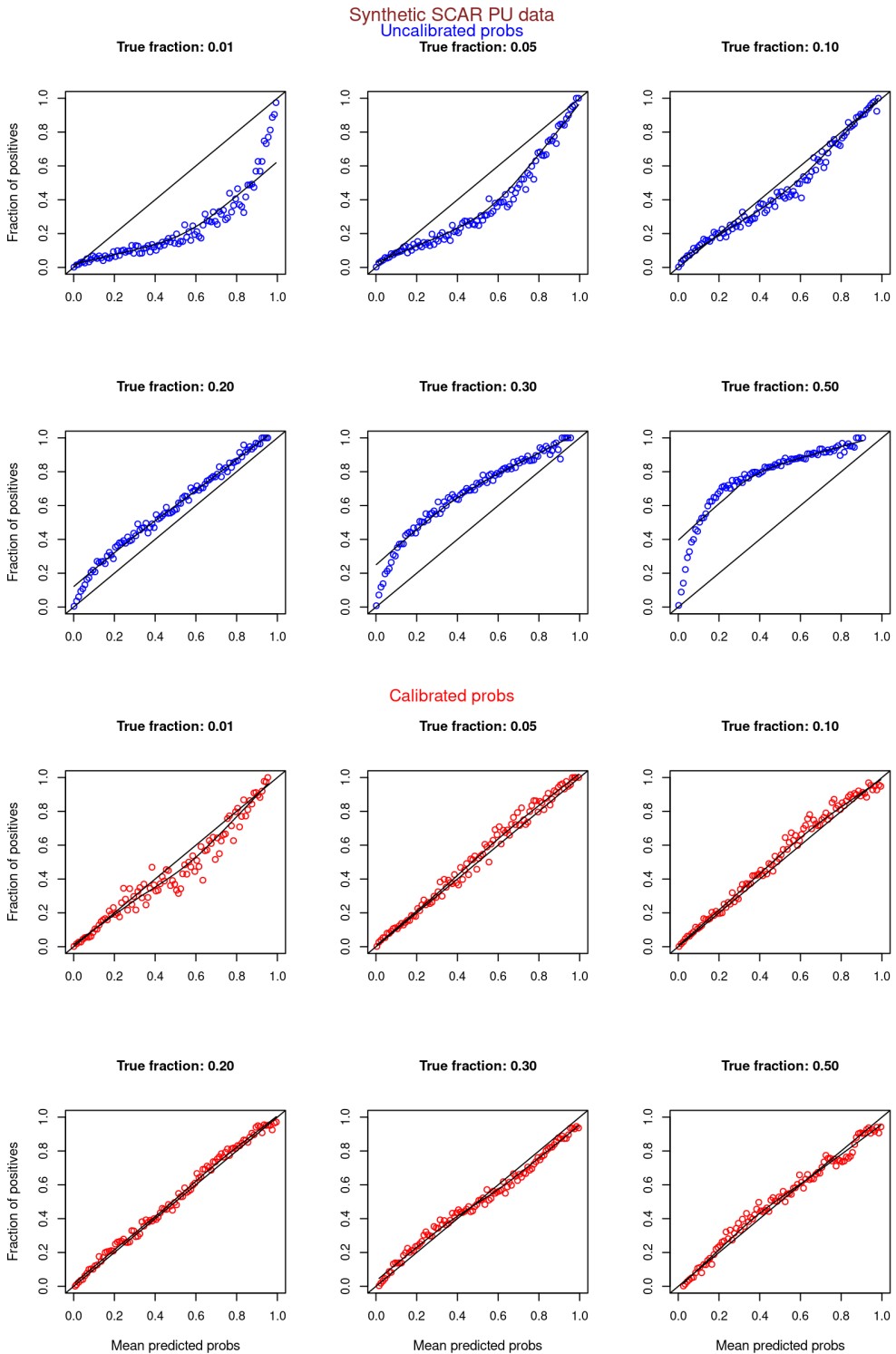

Figure 7: **Calibration curves for Synthetic SCAR datasets (both positive and unlabeled examples)**. Synthetic datasets were generated with different fractions of positives (1%, 5%, 10%, 20%, 30%, and 50%) among the unlabeled examples. class_sep=0.3, number of attributes=100, $|P| = 5,000$ and $|U| = 50,000$. Calibration curves were generated using both positive and unlabeled examples (Uncalibrated probabilities - blue, calibrated probabilities - red).

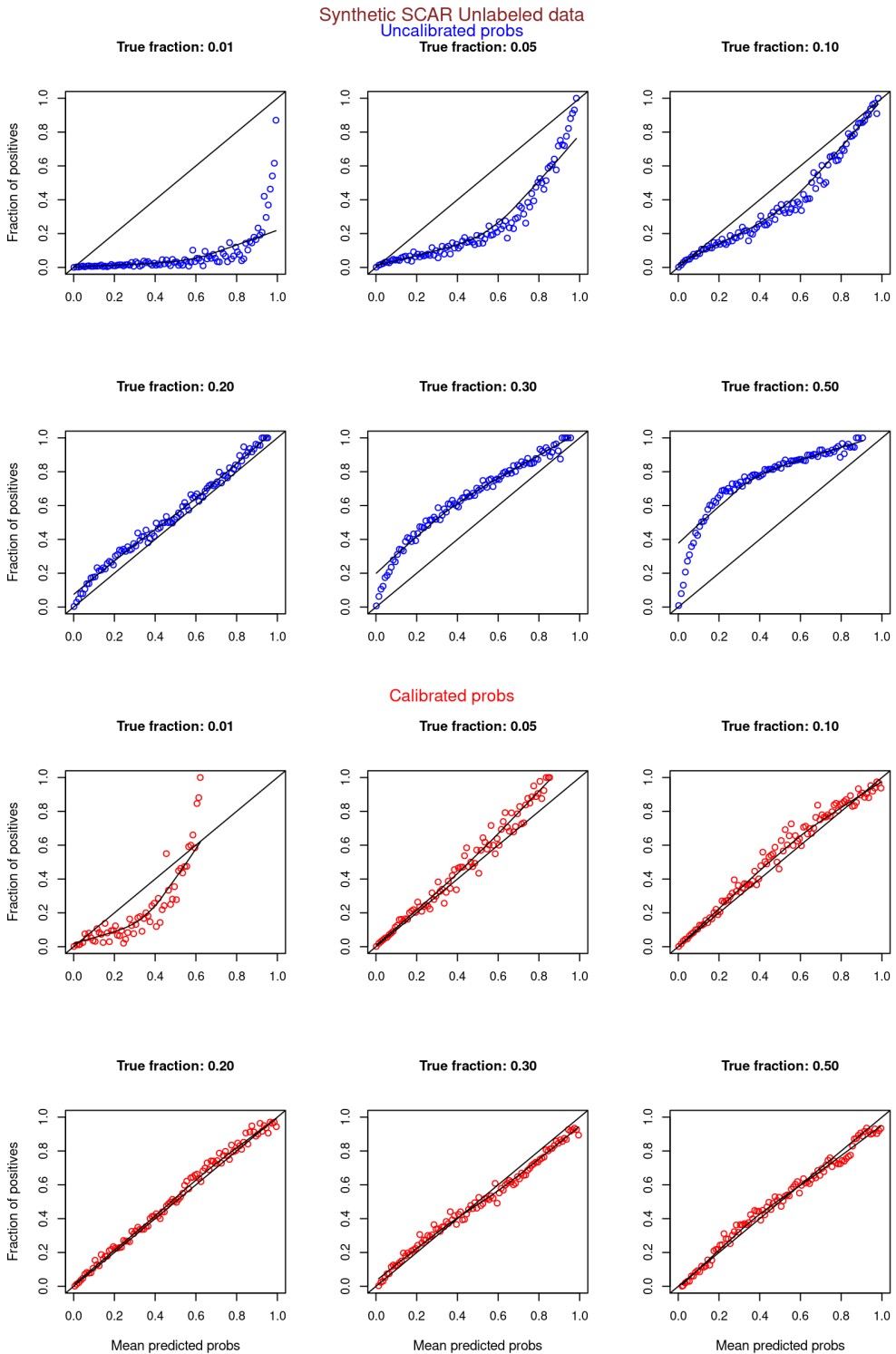

Figure 8: **Calibration curves for Synthetic SCAR datasets (only unlabeled examples)**. Synthetic datasets were generated with different fractions of positives (1%, 5%, 10%, 20%, 30%, and 50%) among the unlabeled examples. class_sep=0.3, number of attributes=100, $|P| = 5,000$ and $|U| = 50,000$. Calibration curves were generated using only unlabeled examples (Uncalibrated probabilities - blue, calibrated probabilities - red).

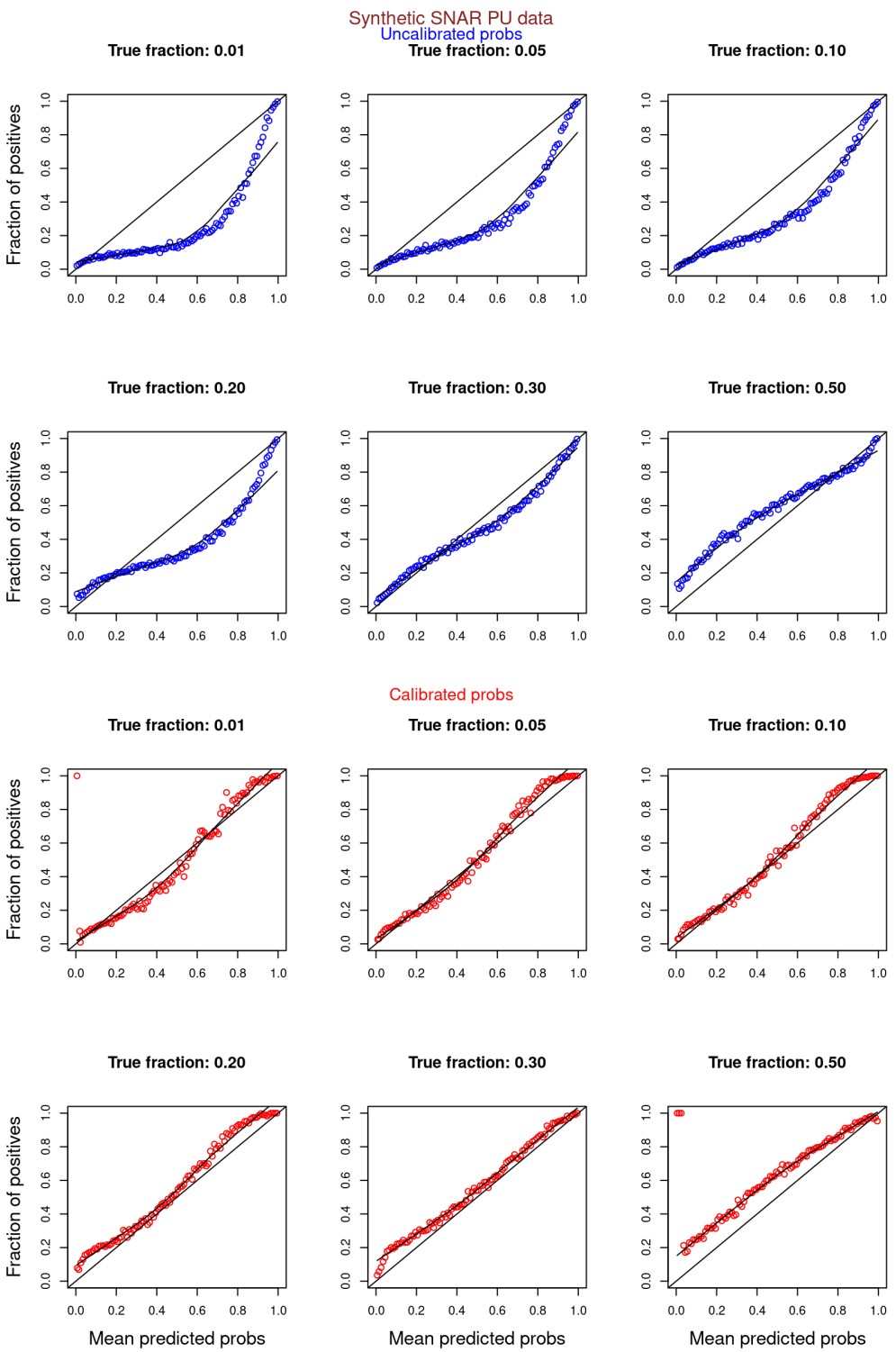

Figure 9: **Calibration curves for Synthetic SNAR datasets (both positive and unlabeled examples)**. Synthetic datasets were generated with different fractions of positives (1%, 5%, 10%, 20%, 30%, and 50%) among the unlabeled examples. class_sep=0.3, number of attributes=100, number of positive subclasses=5, $|P| = 20{,}000$ (4,000 from each subclass) and $|U| = 50{,}000$. Calibration curves were generated using both positive and unlabeled examples (Uncalibrated probabilities - blue, calibrated probabilities - red).

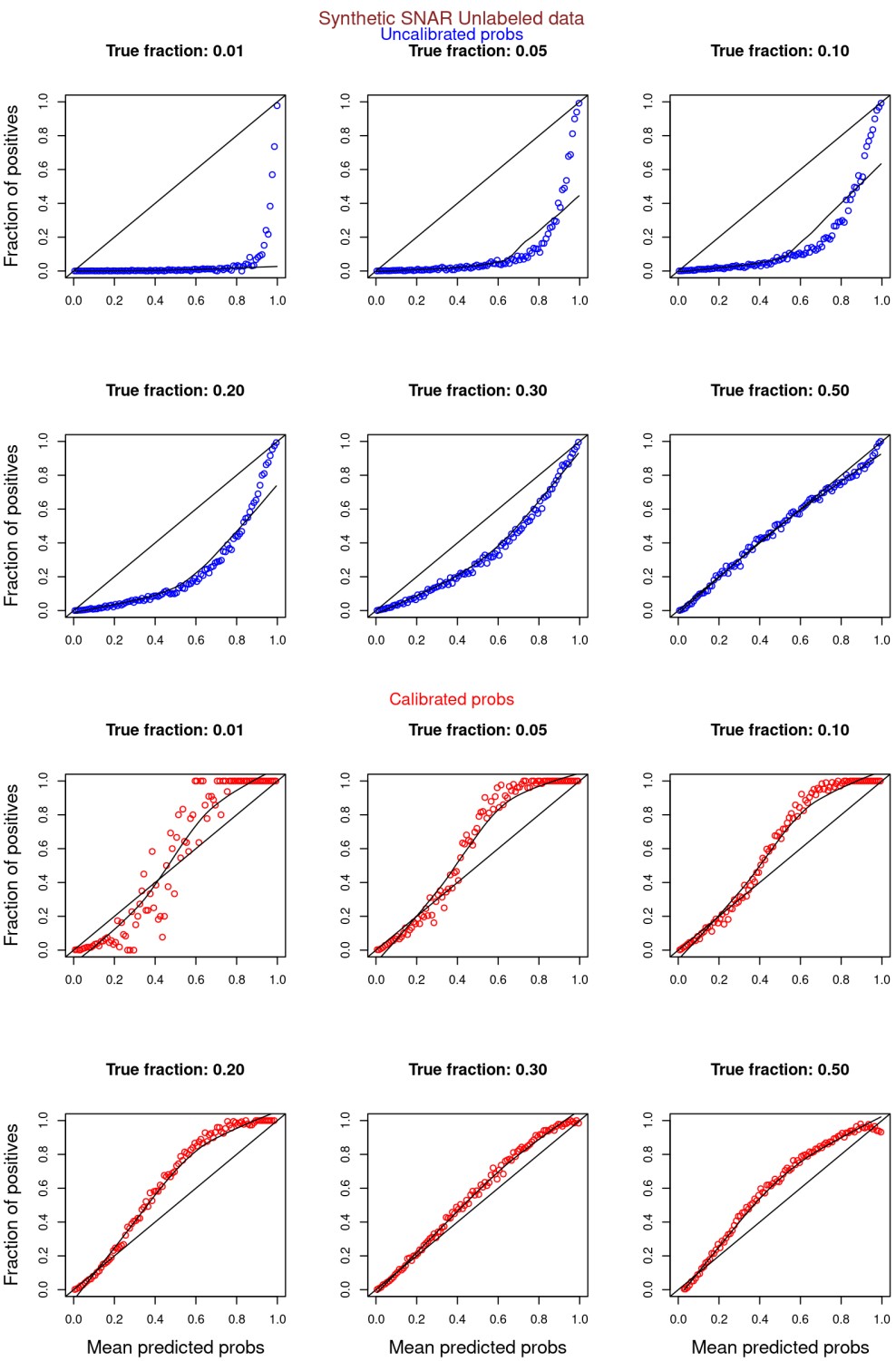

Figure 10: **Calibration curves for Synthetic SNAR datasets (only unlabeled examples)**. Synthetic datasets were generated with different fractions of positives (1%, 5%, 10%, 20%, 30%, and 50%) among the unlabeled examples. class_sep=0.3, number of attributes=100, number of positive subclasses=5, $|P| = 20,000$ (4,000 from each subclass) and $|U| = 50,000$. Calibration curves were generated using only unlabeled examples (Uncalibrated probabilities - blue, calibrated probabilities - red).

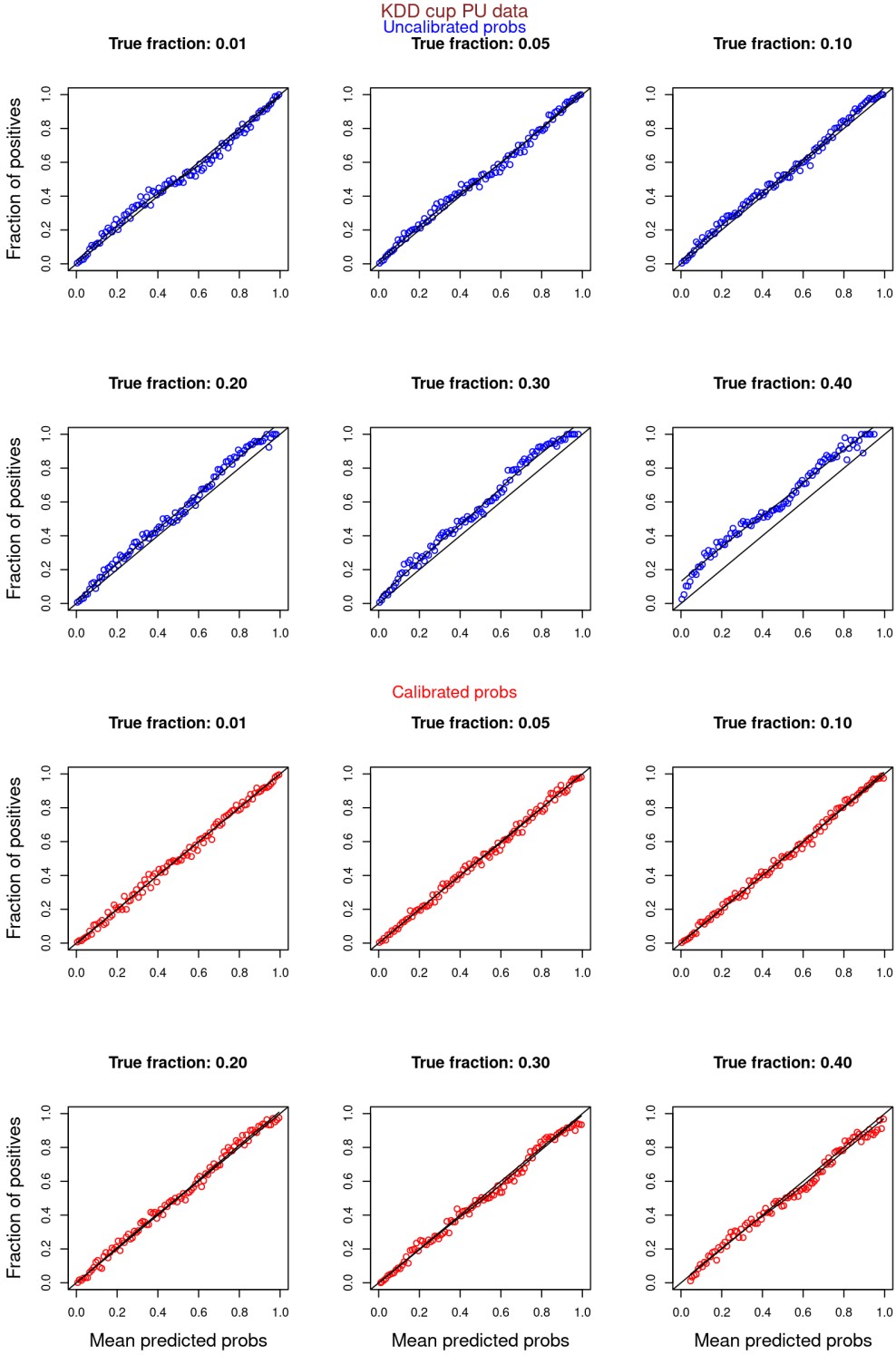

Figure 11: **Calibration curves for SCAR KDD Cup 2004 particle physics dataset (both positive and unlabeled examples)**. Unlabeled sets contained 1%, 5%, 10%, 20%, 30%, and 40% positive examples. Calibration curves were generated using both positive and unlabeled examples (Uncalibrated probabilities - blue, calibrated probabilities - red).

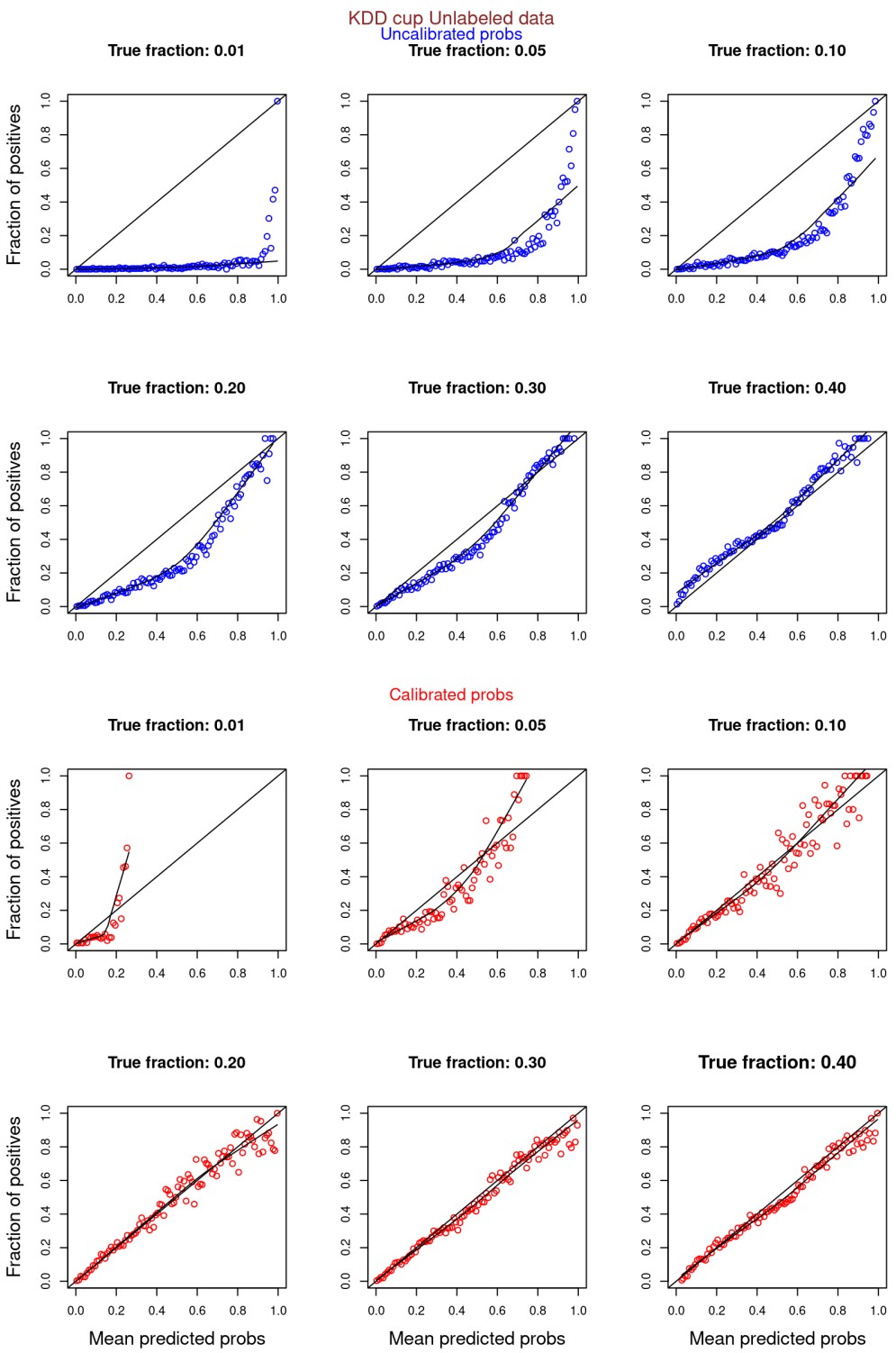

Figure 12: **Calibration curves for SCAR KDD Cup 2004 particle physics dataset (only unlabeled examples)**. Unlabeled sets contained 1%, 5%, 10%, 20%, 30%, and 40% positive examples. Calibration curves were generated using only unlabeled examples (Uncalibrated probabilities - blue, calibrated probabilities - red).

# D    Improving classification performance with PULSCAR and PULSNAR

## D.1    Algorithm for improving classification

Algorithm 5 shows the complete pseudocode to improve classification performance with PULSCAR and PULSNAR. The algorithm returns the following six classification metrics: *Accuracy, AUC-ROC, Brier score (BS), F1, Matthew's correlation coefficient (MCC)*, and *Average precision score (APS)*. The approach to enhancing the classification performance is as follows:

**Using PULSCAR:**  After estimating the $\alpha$, the class 1 predicted probabilities of only unlabeled examples are calibrated using Algorithm 4. The calibrated probabilities of the unlabeled examples are sorted in descending order, and the labels of top $\alpha|U|$ unlabeled examples with the highest calibrated probabilities are flipped from 0 to 1 (probable positives). We then train and test an ML classifier (XGBoost) with 5-fold CV using the labeled positives, probable positives, and the remaining unlabeled examples. The classification performance metrics are calculated using the ML predictions and the true labels of the data.

**Using PULSNAR:**  The PULSNAR algorithm divides the labeled positive examples into $k$ clusters. For each cluster, after estimating $\alpha_j$ for $j$ in $1 \ldots k$, the class 1 predicted probabilities of only unlabeled examples are calibrated using Algorithm 4. Since each unlabeled example has $k$ calibrated probabilities, we compute the final calibrated probability for each unlabeled example using the Formula 6. The final $\alpha$ is calculated by summing the $\alpha_j$ values over the $k$ clusters. The final calibrated probabilities of the unlabeled examples are sorted in descending order, and the labels of top $\alpha|U|$ unlabeled examples with the highest calibrated probabilities are flipped from 0 to 1 (probable positives). We then train and test an ML classifier (XGBoost) with 5-fold CV using the labeled positives, probable positives, and the remaining unlabeled examples. The classification performance metrics are calculated using the ML predictions and the true labels of the data.

---

**Algorithm 5** calculate_classification_metrics

---

**Input**: X ($X_p \cup X_u$), y ($y_p \cup y_u$), y_true, bin_method, n_bins, predicted_probabilities, $\alpha$
**Output**: classification_metrics (accuracy, roc auc, brier score, f1, Matthew's correlation coefficient, average precision)

  1: p $\leftarrow$ predicted_probabilities
  2: $\hat{p} \leftarrow$ calibrate_probabilities(p, y, n_bins, calibration_method, 'U', $\alpha$)
  3: sort $\hat{p}$ in descending order
  4: $\hat{y_u} \leftarrow$ flip labels of top $\alpha|U|$ unlabeled examples with highest $\hat{p}$
  5: y $\leftarrow y_p \cup \hat{y_u}$
  6: predicted_probabilities (p) $\leftarrow \mathcal{A}(X, y)$
  7: **return** accuracy(p, y_true), auc(p, y_true), bs(p, y_true), f1(p, y_true), mcc(p, y_true), aps(p, y_true)

---

## D.2    Experiments and Results

We applied Algorithm 5 to synthetic SCAR and SNAR datasets to get the performance metrics for the XGBoost model with PULSCAR and PULSNAR, respectively. The classification performance metrics were also calculated without applying the PULSCAR or PULSNAR algorithm, in order to determine the improvement in the classification performance of the model. The experiment was repeated 40 times by selecting different train and test sets using 40 random seeds to compute the 95% confidence interval (CI) for the metrics.

Figures 13 and 14 show the classification performance of the XGBoost model with/without the PULSCAR or PULSNAR algorithm on synthetic SCAR and SNAR data, respectively. The classification performance using PULSCAR or PULSNAR increased significantly over XGBoost alone. As the proportion of positives among the unlabeled examples increased, the performance of the model without PULSCAR or PULSNAR (blue) worsened significantly more than when using PULSCAR or PULSNAR.

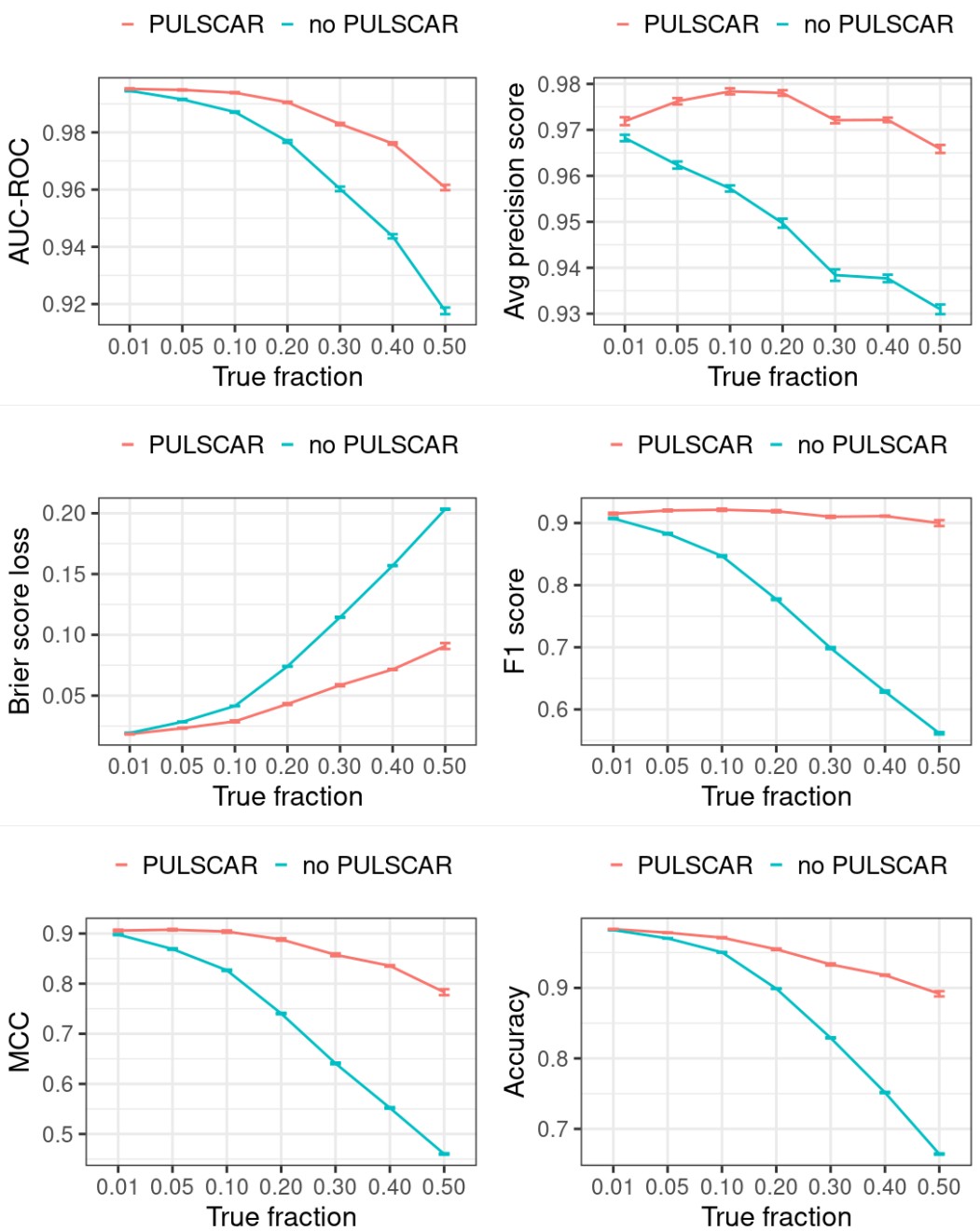

Figure 13: **Classification performance of XGBoost model on synthetic SCAR datasets with and without the PULSCAR algorithm**. Synthetic datasets were generated with different fractions of positives (1%, 5%, 10%, 20%, 30%, 40%, and 50%) among the unlabeled examples. class_sep=0.3, number of attributes=100, $|P| = 5,000$ and $|U| = 50,000$. *"no PULSCAR"* (blue): XGBoost model was trained and tested with 5-fold CV on the given data; the classification metrics were calculated using the model predictions and true labels. *"PULSCAR"* (red): PULSCAR algorithm was used to find the proportion of positives among unlabeled examples ($\alpha$); using $\alpha$, probable positives were identified; XGBoost model was trained and tested with 5-fold CV on labeled positives, probable positives, and the remaining unlabeled examples; classification metrics were calculated using the model predictions and true labels. The error bars represent 95% CIs for the performance metrics.

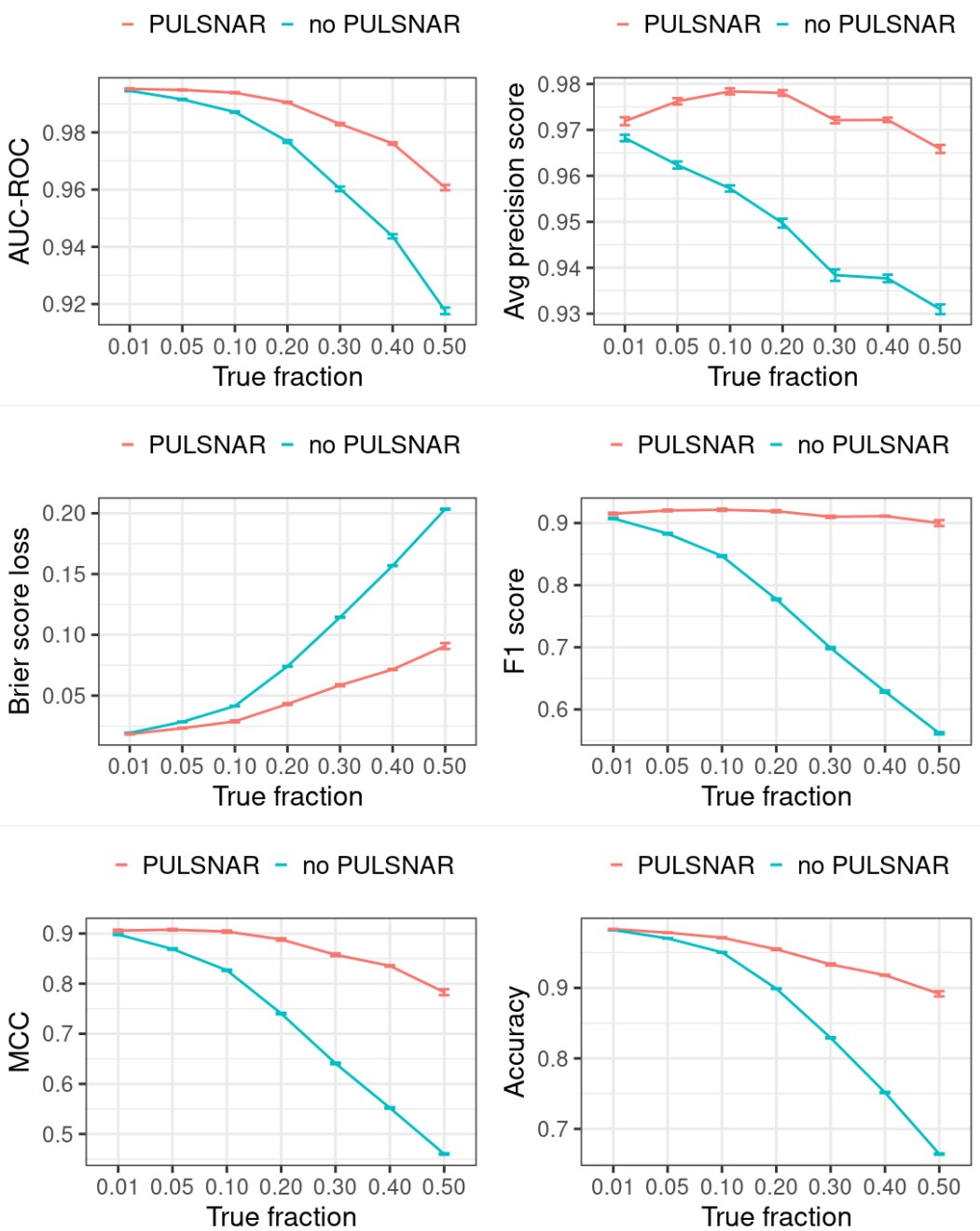

Figure 14: **Classification performance of XGBoost model on synthetic SNAR datasets with and without the PULSNAR algorithm**. Synthetic datasets were generated with different fractions of positives (1%, 5%, 10%, 20%, 30%, 40%, and 50%) among the unlabeled examples. class_sep=0.3, number of attributes=100, number of positive subclasses=5, $|P| = 20,000$ (4,000 from each subclass) and $|U| = 50,000$. *"no PULSNAR"* (blue): XGBoost model was trained and tested with 5-fold CV on the given data; the classification metrics were calculated using the model predictions and true labels. *"PULSNAR"* (red): PULSNAR algorithm was used to find the proportion of positives among unlabeled examples ($\alpha$); using $\alpha$, probable positives were identified; XGBoost model was trained and tested with 5-fold CV on labeled positives, probable positives, and the remaining unlabeled examples; classification metrics were calculated using the model predictions and true labels. The error bars represent 95% CIs for the performance metrics.

# E    DEDPUL vs. PULSNAR: $\alpha$ estimation

Public implementations of the PU learning methods KM1, KM2, and TICE were not scalable; they either failed to execute or would have taken weeks to run the multiple iterations required to obtain confidence estimates for large datasets. We thus could not compare our method with KM1, KM2, and TICE on large datasets and used only DEDPUL for comparison. Importantly, it was previously demonstrated that the DEDPUL method outperformed these three methods on several UCI ML benchmark and synthetic datasets Ivanov (2020).

We compared our algorithm with DEDPUL on synthetic SNAR datasets with different fractions (1%, 5%, 10%, 20%, 30%, 40%, and 50%) of positives among unlabeled examples. In our experiments, we observed that class imbalance (ratio of majority class to minority class) could affect the $\alpha$ estimates. So, we used 4 different sample sizes: 1) positive: 5,000 and unlabeled: 5,000; 2) positive: 5,000 and unlabeled: 25,000; 3) positive: 5,000 and unlabeled: 50,000; 4) positive: 5,000 and unlabeled: 100,000. For each sample size and fraction, we generated 20 datasets using sklearn's *make_classification()* method with random seeds 0-19 to compute 95% CI. We used class_sep=0.3 for each dataset to create difficult classification problems. All datasets were generated with 100 attributes and 6 labels (0-5), defining '0' as negative and 1-5 as positive subclasses. The positive set contained 1000 examples from each positive subclass in all datasets. The unlabeled set comprised k% positive examples with labels (1-5) flipped to 0 and (100-k)% negative examples. The unlabeled positives were markedly SNAR, with the 5 subclasses comprising 1/31, 2/31, 4/31, 8/31, and 16/31 of the unlabeled positives.

Figure 15 shows the $\alpha$ estimates by DEDPUL and PULSNAR on synthetic SNAR data. For smaller true fractions (1%, 5%, 10%), DEDPUL returned close $\alpha$ estimates, but for larger fractions (20%, 30%, 40%, and 50%), it underestimated $\alpha$. Also, as the class imbalance increased, the performance of DEDPUL dropped, especially for larger true fractions. The estimated $\alpha$ by the PULSNAR method was close to the true $\alpha$ for all fractions and sample sizes.

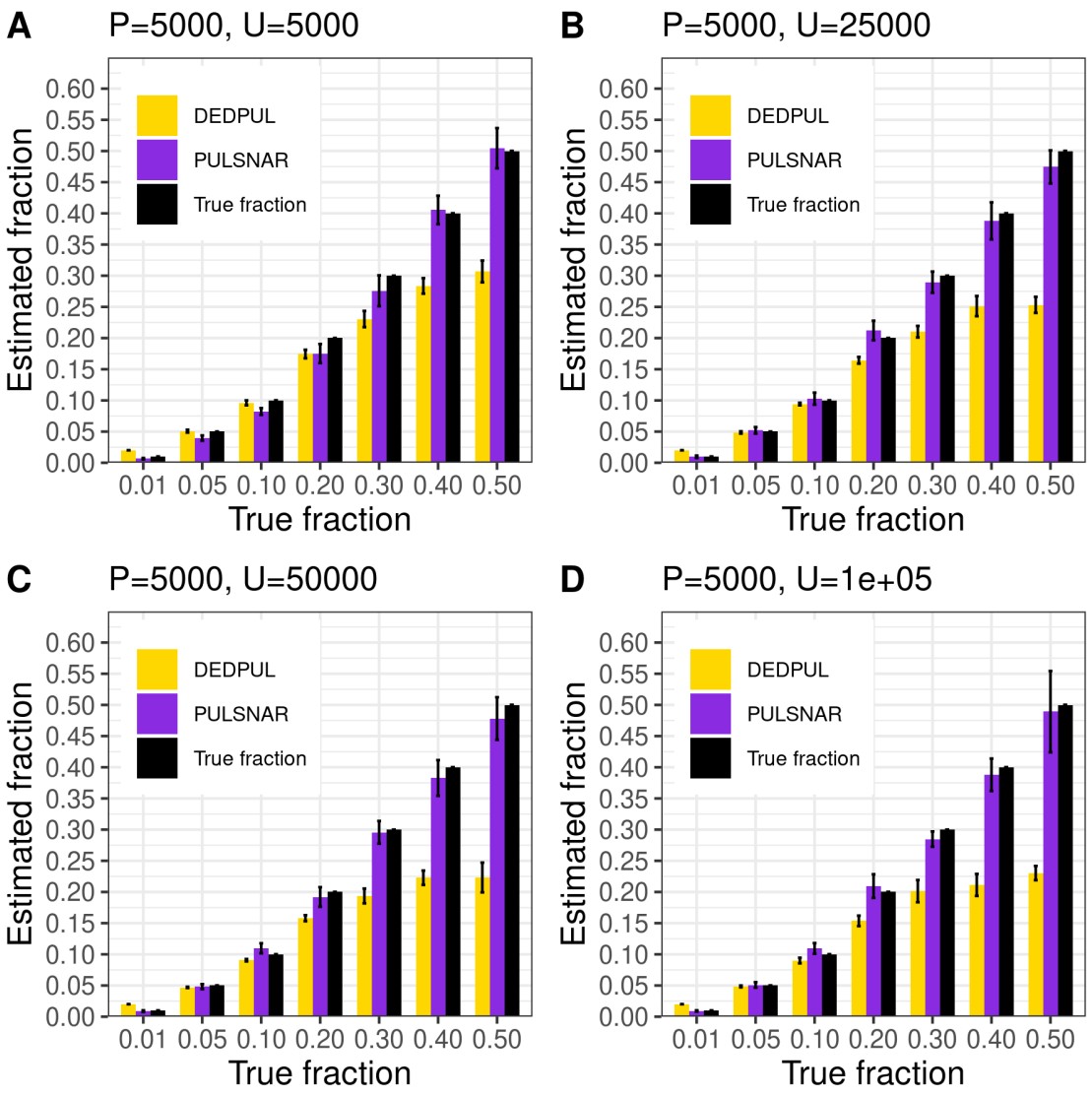

Figure 15: **PULSNAR and DEDPUL evaluated on synthetic SNAR datasets**. The bar represents the mean value of the estimated $\alpha$, with 95% CI for estimated $\alpha$.

