# OpenReview forum: "Positive Unlabeled Learning Selected Not At Random (PULSNAR): class proportion estimation when the SCAR assumption does not hold"
_TMLR — Rejected by TMLR_

### Review · Reviewer_zoS7 · 2023-11-15

**Summary Of Contributions:**

This paper focuses on the class prior estimation problem in PU learning without the selected completely at random (SCAR) assumption. It first utilizes a partial matching strategy to find the class prior under SCAR assumption. To be specific, the paper proposes an upper bound of class prior and utilizes it as the estimation of class prior. Then, the paper employs a divide-and-conquer approach to transform the Selected Not At Random (SNAR) data into several subsets where the data approximates the SCAR assumption. Finally, the paper estimates the class prior in each subset and then utilize the summation of class priors as the estimation of class prior in SNAR data. Several empirical studies are also performed in this paper.

**Audience:**

Yes

**Broader Impact Concerns:**

There is no other concern.

**Claims And Evidence:**

Yes

**Requested Changes:**

1) Why the proposed PULSCAR method outperforms other methods which utilize the same upper bound as the estimation of class prior?
2) The performance of PULSNAR is associated with the performance of PULSCAR in each subproblem. However, roughly utilizing the upper bound as estimation may incur error and the kernel density estimation process can also incur error. How to control the overall error in PULSNAR which utilizes the summation of class priors.
3) Why utilizing the multi-class datasets and select the specific class to generate the PU learning datasets? There are also binary datasets with multiple clusters.
4) Why only performing the experiments on small class prior? The paper should show the performance in a larger range of class prior.

**Strengths And Weaknesses:**

Strength:
1) The paper is well organized, and the idea of the proposed class prior estimation method is easy to follow.
2) The proposed divide-and-conquer approach is interesting and it seems a feasible method to solve the PU learning problem under SNAR assumption.

Weakness:
1) The idea of the class prior estimation under SCAR assumption in this paper has been studied by Kato et al [1], which utilize the same upper bound as the class prior.
2) The proposed PULSNAR method seems can incorporate with any SCAR class prior estimation method, since the proposed PULSCAR is not a new method, the contribution of this paper seems only the proposed divide-and-conquer strategy.
3) The experiments only performed under relatively small class prior which is less than 0.5 and in some data sets the largest class prior is only 0.1, the paper should perform the experiments with larger range of class prior to make the results more persuasive.
4) The experiments of PULNAR only performed on 2 real-world datasets, and the ratio of labeled positive is too large, the paper should perform experiments on more datasets and test the performance of PULNAR with a small number of labeled positive.
5) The solving process of Eq.3 is not clear in the paper, and some of the explanation of pseudo code in Algorithm 1 are missing, such as ``estimation_range’’ and ``f’(\alpha)’’.

[1] Kato, Xu, Niu, and Sugiyama. Alternate estimation of a classifier and the class-prior from positive and unlabeled data. arXiv:1809.05710, 2018.

---

> ### Author Response · Authors · 2023-12-08
> **Response to review of Paper1798 by Reviewer zoS7 (part 1)**
>
> We appreciate the careful review, and respond to the concerns below:
> >Weakness:
> >1. The idea of the class prior estimation under SCAR assumption in this paper has been studied by Kato et al [1], which utilize the same upper bound as the class prior.
> >2. The proposed PULSNAR method seems can incorporate with any SCAR class prior estimation method, since the proposed PULSCAR is not a new method, the contribution of this paper seems only the proposed divide-and-conquer strategy.
>
> Regarding the novelty of our SCAR-based PULSCAR method, we have noted in the paper that other density-based PU techniques [1,2,3] have also leveraged the relationship $\alpha f_p(x) ≤ f_u(x)$ (property 2) for the upper bound. However, our PULSCAR method makes several innovations, including using a beta kernel, introducing an objective function, and determining $\alpha$ through finite differences, as outlined in section 3.2. The fact that it outperformed other SCAR-based methods is also evidence that we made an advance.
> Indeed, if there were a better SCAR-assuming estimator of $\alpha$ than PULSCAR, it could be substituted into PULSNAR and potentially provide better results. However, our comparisons of PULSCAR with other methods that make the SCAR assumption showed it was superior (Figure 3A, 4A, and 4B). Given the problematic runtime performance of other methods (as mentioned in the captions to Figures 4 and 5) and our aim to offer a high-performance Python library compatible with many scikit-learn methods, we avoided creating unmaintainable hybrids.
>
> > 3. The experiments only performed under relatively small class prior which is less than 0.5 and in some data sets the largest class prior is only 0.1, the paper should perform the experiments with larger range of class prior to make the results more persuasive.
>
> Note that we did not evaluate a range of class prior values, we evaluated a range of $\alpha$ values. In PU learning, the class prior is the fraction of positives among both the positive and the unlabeled examples taken together. $\alpha$, as used in some other publications, is the fraction of positives among the unlabeled examples.  We tested PULSCAR and PULSNAR on synthetic datasets, assessing seven $\alpha$ values from 1% to 50%. Our techniques can work above that value (we have tested to 90%, not shown), but competing techniques fail to provide meaningful estimates. We can mention this in the discussion.
>
> Also, the reported prevalence of corrupted labels in real-world datasets varies between 8.0% and 38.5%[4]. Consequently a range from 1-50% should cover most use cases. Sections 4.2 and 4.3 detail how we add positive examples to the unlabeled set. In order to mimic real-world scenarios, we designated the class with fewer instances as positive and the class with more instances as unlabeled. Given the limited sample size of positive instances in machine learning benchmark datasets, opting for large $\alpha$ values would result in very few or no positive examples for testing. Hence, we were unable to evaluate our algorithms using large $\alpha$ values on those datasets.
>
>
> >4. The experiments of PULNAR only performed on 2 real-world datasets, and the ratio of labeled positive is too large, the paper should perform experiments on more datasets and test the performance of PULNAR with a small number of labeled positive.
>
> Can you please clarify "the ratio of labeled positive is too large"? That is, the ratio of what to what, and why that ratio is too large? Figures 4 and 5 captions note the inability to run other PU methods (KM1, KM2, TICE, DEDPUL) on large benchmark datasets, preventing a more thorough comparison. Consequently, we included only the datasets where comparisons with other methods were feasible. In terms of tests on a small number of positives, we did test PULSNAR with 2000 positive examples in the synthetic data where each subclass had 400 instances. As one approaches miniscule sample sizes, all methods will fail -- our focus was to compare among methods.
>
> References:
> 1. Dmitry Ivanov. Dedpul: Difference-of-estimated-densities-based positive-unlabeled learning. In 2020 19th IEEE International Conference on Machine Learning and Applications (ICMLA), pp. 782–790. IEEE, 2020.
> 2. Kato, Xu, Niu, and Sugiyama. Alternate estimation of a classifier and the class-prior from positive and unlabeled data. arXiv:1809.05710, 2018.
> 3. Saurabh Garg, Yifan Wu, Alexander J Smola, Sivaraman Balakrishnan, and Zachary Lipton. Mixture proportion estimation and pu learning: a modern approach. Advances in Neural Information Processing Systems, 34:8532–8544, 2021.
> 4. Song H, Kim M, Park D, Shin Y, Lee JG. Learning from noisy labels with deep neural networks: A survey. IEEE Transactions on Neural Networks and Learning Systems. 2022 Mar 7.

---

> ### Author Response · Authors · 2023-12-08
> **Response to review of Paper1798 by Reviewer zoS7 (part 2)**
>
> >5. The solving process of Eq.3 is not clear in the paper, and some of the explanation of pseudo code in Algorithm 1 are missing, such as estimation_range’’ and f’(\alpha)’’.
>
> We would be happy to provide further clarification of Algorithm 1 in a revision, including estimation_range and $f’(\alpha)$. Briefly, the range for the $\alpha$ estimate is defined by the estimation_range in Algorithm 1, with minimum and maximum values of 0 and 1, respectively. The function $f(\alpha)$ in Algorithm 1, line 15, corresponds to Equation 3. In line 16, we use finite differences to find the slope of $f(\alpha)$, which is $f’(\alpha)$ within the estimation_range [0, 1]. Subsequently, on line 17, we identify the index where $f’(\alpha)$ exhibits the maximum change, determining the value of $\alpha$.
>
>
>
> >Requested Changes:
> >1. Why the proposed PULSCAR method outperforms other methods which utilize the same upper bound as the estimation of class prior?
>
> Section 6 (Discussion and Conclusion) elaborates why our method outperformed other methods, and that the KM1, KM2, and TiCE algorithms exhibited scalability issues and could not process large datasets with high dimensions. We noted that it is plausible that their inherent limitations with data size and high dimension contributed to inaccurate $\alpha$ estimates for some of our test datasets. Also, we explained that DEDPUL, which also utilizes probability density for $\alpha$ estimation, employs a Gaussian kernel, a choice we identified as problematic (as detailed in section 3.3). Furthermore, we stated that we believe our robust approach to density-based $\alpha$ estimation using Equation 3 may have more reliable convergence properties than the EM algorithm used by DEDPUL. We think these explanations cover why our approach outperforms others, and that there are not additional reasons.
>
>
> >2. The performance of PULSNAR is associated with the performance of PULSCAR in each subproblem. However, roughly utilizing the upper bound as estimation may incur error and the kernel density estimation process can also incur error. How to control the overall error in PULSNAR which utilizes the summation of class priors.
>
> In the "Discussion and Conclusion" section (Section 6), we noted that an earlier version of our PULSCAR utilized a Gaussian kernel. With the Gaussian kernel, we observed that PDF values of positive and unlabeled examples had substantial errors, sometimes resulting in the combined cluster $\alpha$ sums exceeding 1 for PULSNAR. As described in 3.3, Gaussian boundary biases and errors in polynomial approximations were overcome by implementing a beta kernel estimator, which generated PDF values bounded within the [0 . . . 1] range for positive and unlabeled examples. Given our errors in $\alpha$ are smaller than all compared methods, we believe we have advanced error control. Additional improvements would be the focus of future studies.
>
>
> >3. Why utilizing the multi-class datasets and select the specific class to generate the PU learning datasets? There are also binary datasets with multiple clusters.
>
> It was convenient with synthetic data generation to create sub-types of positives using multiple classes via sklearn make_classification() as described in 4.1. This allowed us to know the exact sub-type classification and sample different proportions from different subclasses to create reliable SNAR simulations with known answers. With real-world binary data with multiple clusters there would not necessarily be a well-defined sub-typing procedure, given many clusters are possible. We thus used multi-class benchmark data in a similar fashion as we did for the simulated data.
>
> >4. Why only performing the experiments on small class prior? The paper should show the performance in a larger range of class prior.
>
> Please see the response to weakness 3 above.

---

### Review · Reviewer_Eoyw · 2023-12-04

**Summary Of Contributions:**

In this paper, the authors propose two approaches to estimate positive proportion in PU learning under the SCAR and the SNAR settings respectively. An interesting objective function is constructed to obtain the estimated positive proportion.

**Audience:**

Yes

**Claims And Evidence:**

No

**Requested Changes:**

1. The pseudo-codes are rather informal. It is suggested to rewrite them in a more formal and clearer way.
2. The proposed approach is agnostic to the used binary classifier. However, only the XGBoost is tested in the paper. It is desirable to show the proposed approaches work well with other classifiers.
3. The so-called 'real-world' SNAR datasets are actually synthetic datasets. It is desirable to test the SNAR setting on some REAL real-world datasets.
4. In Figure 3 A and Figure 4 A, it seems that the existing approach DEDPUL can estimate the positive proportion more accurately than the proposed approach. Please elaborate possible reasons.

**Strengths And Weaknesses:**

Strengths:
* The topic investigated in the paper is interesting and it is meaningful to study the more realistic SNAR setting for PU learning.
* Experiments are conducted on synthetic and real-world datasets to show the effectiveness of proposed approaches.

Weaknesses:
* Details of proposed approaches are not presented clearly. For example, why not simply take the minimum point of Eq. (3) as the estimated positive proportion? And what is the motivation for weighting features before clustering in PULSNAR? These implementation details need further clarification, and their effectiveness should be empirically demonstrated.
* It makes me confused that the proposed approaches induce PDF of positive class directly from labeled positive examples. In SCAR setting, there are potential positive examples in unlabeled data. While in SNAR setting, there are also positive examples in other clusters of labeled positive examples. Why can these positive examples be neglected when inducing the PDF. It is necessary to show such an implementation is reasonably. At least, empirical studies can be conducted to show the approximation ability of the induced PDF.
* Experiments are not comprehensive enough. The newest comparing approach is proposed in 2021. And it is necessary to show that the proposed approach can lead to better classification performance than existing approach (e.g. DEDPUL). There is no such experiment in current version.

---

> ### Author Response · Authors · 2023-12-19
> **Response to review of Paper1798 by Reviewer Eoyw (part 1)**
>
> We appreciate the careful review, and respond to the concerns below:
>
> >Details of proposed approaches are not presented clearly. For example, why not simply take the minimum point of Eq. (3) as the estimated positive proportion? And what is the motivation for weighting features before clustering in PULSNAR? These implementation details need further clarification, and their effectiveness should be empirically demonstrated.
>
> Given the non-convex nature of our objective function (see section 3.2), the presence of multiple minima is anticipated. The real challenge in such a scenario is identifying the minimum that accurately represents the $\alpha$ estimate. Therefore, instead of looking for the minimum value, our approach finds the point of greatest divergence, characterized by the maximal change in the slope of the objective function (refer to line numbers 16 and 17 of Algorithm 1). This corresponds to how the logarithm steeply approaches negative infinity as  $\alpha f_p(x)$ approaches $ f_u(x)$. This point of greatest divergence represents the point where the density of the positive examples begins to exceed the density of the unlabeled examples.
>
> We select all important features along with their contributions (represented by gain scores in the case of XGBoost) utilized by the classifier to learn from the data. Features with higher gain scores are deemed more important. By scaling those important features by their gain scores, we assign higher weights to the more informative features and lower weights to redundant ones. This approach helps the clustering algorithm make clear distinctions between examples, producing cleaner clusters, each predominantly containing examples from a single class. In our experiments (results now shown in the paper), when we did not scale important features by their gain scores, the resulting clusters included mixtures of examples from all classes, leading to inaccurate $\alpha$ estimation by PULSNAR.
>
> In order to fit the key content of the paper within 12 pages, we skipped results based on experiments without weighting features, but will mention that these experiments were done in the revision.
>
> >It makes me confused that the proposed approaches induce PDF of positive class directly from labeled positive examples. In SCAR setting, there are potential positive examples in unlabeled data. While in SNAR setting, there are also positive examples in other clusters of labeled positive examples. Why can these positive examples be neglected when inducing the PDF. It is necessary to show such an implementation is reasonably. At least, empirical studies can be conducted to show the approximation ability of the induced PDF.
>
> In both SCAR and SNAR scenarios, a certain proportion ($\alpha$) of positive examples exists within the pool of unlabeled examples. Our PULSCAR and PULSNAR methods aim to estimate this unknown $\alpha$. Without prior knowledge of positive examples in the unlabeled set, how can those examples be included to calculate the probability density function (PDF) of positives?
>
> In the SCAR assumption, where a labeled positive is considered an independent and identically distributed (i.i.d) example from the positive distribution, the PDF of unlabeled positives closely resembles that of labeled positives. Leveraging this property of the SCAR assumption, we seek to estimate $\alpha$ such that subtracting $\alpha f_p(x)$ from $f_u(x)$ matches the resulting PDF with the PDF of true negatives in the unlabeled set. We validated this approach empirically, and Figures 1(A, B, C) illustrate the patterns of various PDFs in cases of correct estimation, overestimation, and underestimation.
>
> >Experiments are not comprehensive enough. The newest comparing approach is proposed in 2021. And it is necessary to show that the proposed approach can lead to better classification performance than existing approach (e.g. DEDPUL). There is no such experiment in current version.
>
> In this paper, our primary emphasis was comparing the $\alpha$ estimates provided by various PU methods. Additionally, the KM1, KM2, and TiCE methods do not provide output probabilities to support classification, preventing comparison of classification performance with these methods. In response to the reviewer's excellent suggestion, we will compare the classification performance between our methods and DEDPUL and present the results in the appendix.

---

> ### Author Response · Authors · 2023-12-19
> **Response to review of Paper1798 by Reviewer Eoyw (part 2)**
>
> >Requested Changes:
> > - The pseudo-codes are rather informal. It is suggested to rewrite them in a more formal and clearer way.
>
> We would be happy to provide clearer pseudo-code in the revision.
>
> >The proposed approach is agnostic to the used binary classifier. However, only the XGBoost is tested in the paper. It is desirable to show the proposed approaches work well with other classifiers.
>
> Our current implementation accommodates three classifiers: XGBoost, Logistic Regression (LR), and CatBoost. We tested all these 3 classifiers for PULSCAR and PULSNAR, but presented results based on only XGBoost in the paper due to the following reasons:
>
> 1. In our experiments, the $\alpha$ estimates derived using XGBoost predictions outperformed those obtained using LR and CatBoost predictions.
> 2. We scale important features by their importance scores to cluster labeled positive examples. The clustering outcomes were better—each cluster predominantly contained examples from a single class—when we applied weights derived from the importance scores of the XGBoost model compared to weighting features using the importance scores from LR or CatBoost. Better clustering improves the accuracy of $\alpha$ estimates by PULSNAR.
> 3. Additionally, despite employing multiple CPU cores, LR and CatBoost exhibited slower performance on large datasets compared to XGBoost.
>
> It's important to note that the objective of this paper is not to compare the performance of different binary classifiers; rather, it aims to introduce empirical algorithms capable of leveraging classifier predictions to estimate $\alpha$. So, selecting a high-performance classifier is always desirable for better $\alpha$ estimates.
>
> We will revise the paper to clarify that while we investigated all three classifiers, and that the results presented are based solely on XGBoost due to the reasons mentioned above.
>
> >The so-called 'real-world' SNAR datasets are actually synthetic datasets. It is desirable to test the SNAR setting on some REAL real-world datasets.
>
> We are unaware of “REAL” benchmark datasets available in the public domain, and they must be constructed through manual assessment of ground-truth -- in which case the unlabeled become labeled. This is why we took an approach of converting “REAL” labeled benchmarks to have unlabeled values. It also allows us to have a ground truth to evaluate methods with.  If you are aware of published SNAR and SCAR benchmark datasets, we would be grateful to learn of them.
>
> We do have poster presentations on patient data where we have performed chart-review on unlabeled patients to confirm our $\alpha$ estimates, and not only find good agreement, but also found the methods could handle sample sizes over 1M. This work will take considerable effort to bring to fruition, preventing their inclusion in the manuscript, but we are happy to cite these posters once the double-blind review phase concludes. It should be noted that every other PU-learning publication of which we are aware uses benchmark data where labeled observations are transformed to unlabeled, as in our work.
>
> > In Figure 3 A and Figure 4 A, it seems that the existing approach DEDPUL can estimate the positive proportion more accurately than the proposed approach. Please elaborate possible reasons.
>
> In 3A and 4A, depending on the $\alpha$, sometimes PULSCAR has a slight edge, and sometimes DEDPUL does -- however, these appear to be within the noise of the experiments where the 95% confidence intervals overlap. These are datasets where the classes are roughly balanced and the data are SCAR. These methods both have similar theoretical foundations and one would expect comparable performance as noted in the paper -- though with small $\alpha$, DEDPUL does appear to have large overestimates (e.g. it gave 2% for 1% true $\alpha$ on both simulated and KDDCup SCAR data). Importantly PULSNAR significantly outperforms all of the methods, including DEDPUL on SNAR data, see Figures 3B, 5, and 15. It should be noted that the performance of DEDPUL experienced a decline when class imbalance increased, and datasets did not adhere to the SCAR assumption (see Figure 15).

---

> > ### Comment · Reviewer_Eoyw · 2024-01-07
> >
> > Thanks for authors' detailed responses. Most of my concerns have been eliminated. I wish authors will report additional experimental results which are promised in their responses in the final version.

---

> ### Author Response · Authors · 2024-01-08
>
> Thank you, we will provide these in a final version, pending a decision. Or are you asking we append these results to the discussion?

---

> > ### Author Response · Authors · 2024-01-11
> >
> > Per your suggestion, we compared the classification performance of DEDPUL and PULSNAR using synthetic datasets with different fractions (1%, 5%, 10%, 20%, 30%, 40%, and 50%) of positive instances among unlabeled examples. Our experiments involved 20,000 positive examples (4,000 from each subclass) and 50,000 unlabeled examples. The classification performance metrics shown in the following table represent the mean of 3 iterations for both DEDPUL and PULSNAR. In the Appendix of the revised manuscript, we will present the mean values of classification performance metrics along with their 95% confidence intervals based on 40 iterations. The table shows that PULSNAR demonstrated superior classification performance compared to DEDPUL on SNAR datasets, particularly when the true alpha value is higher. [MCC: Matthews correlation coefficient, APS: Average precision score, BS: Brier score]
> >
> >
> > | True fraction |    |AUC|     |     APS    |      |      BS   |      |      F1   |     |      MCC   |  |      Accuracy   |
> > |---------------|--------|---------|--------|---------|--------|---------|--------|---------|--------|---------|----------|---------|
> > |               | DEDPUL | PULSNAR | DEDPUL | PULSNAR | DEDPUL | PULSNAR | DEDPUL | PULSNAR | DEDPUL | PULSNAR | DEDPUL   | PULSNAR |
> > | 0.5           | \| 0.382  | 0.974 \| | 0.559  | 0.985 \| | 0.322  | 0.085 \| | 0.384  | 0.929 \|  | -0.155 | 0.812  \| | 0.387    | 0.911 \|  |
> > | 0.4           | \| 0.547  | 0.981 \| | 0.599  | 0.986 \| | 0.275  | 0.074 \| | 0.448  | 0.932 \|  | 0.063  | 0.847  \| | 0.504    | 0.924 \|  |
> > | 0.3           | \| 0.685  | 0.984 \| | 0.651  | 0.985 \| | 0.230  | 0.066 \| | 0.539  | 0.933 \|  | 0.248  | 0.869  \| | 0.617    | 0.934 \|  |
> > | 0.2           | \| 0.847  | 0.986 \| | 0.798  | 0.982 \| | 0.174  | 0.060 \| | 0.636  | 0.928 \|  | 0.484  | 0.877  \| | 0.746    | 0.940 \|  |
> > | 0.1           | \| 0.946  | 0.987 \| | 0.917  | 0.980 \| | 0.114  | 0.055 \| | 0.758  | 0.924 \|  | 0.685  | 0.883  \| | 0.855    | 0.947 \|  |
> > | 0.05          | \| 0.973  | 0.987 \| | 0.954  | 0.977 \| | 0.082  | 0.054 \| | 0.825  | 0.919 \|  | 0.775  | 0.881  \| | 0.902    | 0.948 \|  |
> > | 0.01          | \| 0.984  | 0.988 \| | 0.969  | 0.975 \| | 0.060  | 0.052 \| | 0.873  | 0.916 \|  | 0.835  | 0.881  \| | 0.932    | 0.951 \|  |

---

### Review · Reviewer_Wz6k · 2023-12-06

**Summary Of Contributions:**

The paper delves into Positive and Unlabeled (PU) learning. In this approach, a machine learning algorithm distinguishes between labeled positive instances and a mixture of both positive and negative instances that are unlabeled. Many PU learning algorithms assume that positive instances are selected entirely at random (SCAR), irrespective of their features. However, this assumption doesn't always hold in real-world applications, leading to potential inaccuracies in estimating the proportion of positives among the unlabeled samples and resulting in poor model calibration. To tackle this challenge, the authors introduce two PU learning algorithms: i) PULSCAR (Positive Unlabeled Learning Selected Completely At Random) and ii) PULSNAR (Positive Unlabeled Learning Selected Not At Random). PULSNAR employs a divide-and-conquer approach, leveraging PULSCAR to solve various sub-problems. Experimental results indicate that PULSNAR outperforms state-of-the-art methods on both synthetic and real-world benchmark datasets.

**Audience:**

Yes

**Claims And Evidence:**

Yes

**Requested Changes:**

See Weaknesses

**Strengths And Weaknesses:**

Strength

1. One of the strengths of this paper lies in its exploration of the Selected Not At Random (SNAR) setting within the context of Positive and Unlabeled (PU) learning. By recognizing that positives may not always be selected randomly in real-world scenarios, the authors have introduced a more realistic and nuanced approach through the PULSNAR algorithm.

2. The proposed approach is simple and flexible. By designing a method that can seamlessly nest within existing SCAR-based PU learning algorithms, the authors have ensured that their approach can be integrated with established techniques.

Weakness

1. The objective function designed in Equation 3 leverages f_u\left(x\right) as an upper bound for \alpha f_p\left(x\right). However, the tightness of this upper bound is not guaranteed, which could potentially lead to an inaccurate estimation of \alpha. It would be beneficial to include a discussion or a quantification of the bound tightness to assess the potential impact on the accuracy of \alpha estimation.

2. In Section 3.3, several concepts are introduced without adequate explanation. The terms 'Kernel Bandwidth' and 'Bin Count' are used without clear definitions or context. Furthermore, the details of certain methods, such as those used to determine the number of histogram bins and the specifics of the differential_evolution() optimizer, are not sufficiently elaborated.

3. The methodology of the article proposes to decompose the SNAR problem into multiple SCAR problems via a clustering approach. However, there is no guarantee that after decomposition, each cluster will conform to the SCAR assumption. Even if every cluster were to satisfy the SCAR conditions, an excessive number of clusters could result in a very small number of samples within each cluster. This sparse distribution of data might render the SCAR methods ineffective when applied to individual clusters.

4. In Section 3.4.2, the strategy of iterating n_components over a range of 1 to 25 to compute the Bayesian information criterion is rather brute-force and lacks finesse. Such a method is not only inelegant but can also become highly time-consuming, especially when dealing with large-scale datasets. Moreover, limiting the search to 25 clusters is arbitrary and may not be appropriate for scenarios requiring a higher number of clusters. Relying on this fixed range diminishes the method's adaptability to diverse datasets and scenarios, potentially compromising its effectiveness and efficiency.

---

> ### Author Response · Authors · 2023-12-19
> **Response to review of Paper1798 by Reviewer Wz6k (part 1)**
>
> We appreciate the careful review, and respond to the concerns below:
>
> >The objective function designed in Equation 3 leverages $f_u\left(x\right)$ as an upper bound for $\alpha f_p\left(x\right)$. However, the tightness of this upper bound is not guaranteed, which could potentially lead to an inaccurate estimation of $\alpha$. It would be beneficial to include a discussion or a quantification of the bound tightness to assess the potential impact on the accuracy of $\alpha$ estimation.
>
> We appreciate the reviewer's point that the tightness of the upper bound, $ f_u(x)$, for $\alpha f_p(x)$ in the PULSCAR algorithm is not guaranteed. It's a challenging aspect that has not been theoretically resolved in our current work, nor in prior literature. Our empirical results suggest greater overestimation as a percentage of ground truth as $\alpha$ gets smaller, which may be a function of limited sample size. We propose adding a point of discussion to this effect in a minor revision.
>
>
> >In Section 3.3, several concepts are introduced without adequate explanation. The terms 'Kernel Bandwidth' and 'Bin Count' are used without clear definitions or context. Furthermore, the details of certain methods, such as those used to determine the number of histogram bins and the specifics of the differential_evolution() optimizer, are not sufficiently elaborated.
>
> In order to fit the key content of the paper within 12 pages, we referenced the approaches incorporated into our algorithms. We would be happy to include additional details about 'Kernel Bandwidth', 'Bin Count', and 'differential_evolution() optimizer' either in the main section or in the appendix. Briefly, we describe ‘Kernel Bandwidth’, ‘Bin Count’, and differential_evolution():
> - Kernel Bandwidth: The kernel bandwidth is the smoothing parameter in kernel density estimation. It influences the overall shape of the estimated distribution curve: a broader bandwidth leads to a smoother and more generalized curve, while a narrower bandwidth creates a more fine-grained curve. As explained in section 3.3, the choice of bandwidth can impact $\alpha$ estimates.
> - Bin Count: Bin count is a parameter that we use to compute the histogram density and beta kernel density using the given predictions from the classification model. As outlined in section 3.3.1, we employ one of the following methods to determine the number of bins for the given data based on the predicted probabilities of positive and unlabeled examples: square root, Sturges’ rule, Rice’s rule, Scott’s rule, and Freedman–Diaconis (FD). We will provide the formula for these methods in the revision.
> - differential_evolution() optimizer: We employ the differential_evolution() function from the SciPy library to determine the kernel bandwidth. It is an optimization algorithm designed to find the global minimum of a multivariate function without requiring gradient information. In our algorithm, it explores the specified bandwidth range, aiming to identify the global minimum of the mean squared error between the histogram and beta kernel densities calculated using predictions from the classifier.
>
> >The methodology of the article proposes to decompose the SNAR problem into multiple SCAR problems via a clustering approach. However, there is no guarantee that after decomposition, each cluster will conform to the SCAR assumption. Even if every cluster were to satisfy the SCAR conditions, an excessive number of clusters could result in a very small number of samples within each cluster. This sparse distribution of data might render the SCAR methods ineffective when applied to individual clusters.
>
> We acknowledge that the subproblems generated by the clustering method may not fully satisfy the SCAR assumption. However, we did not assert or make any guarantee that clusters will satisfy the SCAR assumption. In section 3.4.1, we explained why clustering positive examples converts a SNAR problem into multiple subproblems that are more likely to follow the SCAR assumption.
>
> While the reviewer's concern regarding a small number of positives within each cluster due to excessive clusters, leading to the ineffectiveness PULSNAR, appears valid, we did not observe the 'Knee point detection in BIC' [2] method estimating an excessive number of clusters in any of our experiments. Additionally, as detailed in section 6, our preliminary findings (not presented in the paper) indicate that the PULSNAR $\alpha$ estimation remains robust even when there is an overestimation of the number of clusters in SNAR data.
>
> To mitigate a sparse distribution affecting the PULSCAR method within individual clusters, such that there are few positives per cluster relative to the large set of unlabeled examples, we scale the weight of positive examples using the 'scale_pos_weight' parameter of XGBoost in the classification step. The weight of positive examples is scaled by a factor of k, where $k = \frac{|unlabeled|}{|positive|}$, as explained in section 4.

---

> ### Author Response · Authors · 2023-12-19
> **Response to review of Paper1798 by Reviewer Wz6k (part 2)**
>
> > In Section 3.4.2, the strategy of iterating n_components over a range of 1 to 25 to compute the Bayesian information criterion is rather brute-force and lacks finesse. Such a method is not only inelegant but can also become highly time-consuming, especially when dealing with large-scale datasets. Moreover, limiting the search to 25 clusters is arbitrary and may not be appropriate for scenarios requiring a higher number of clusters. Relying on this fixed range diminishes the method's adaptability to diverse datasets and scenarios, potentially compromising its effectiveness and efficiency.
>
>
> Determining the "optimal number of clusters" is a challenging problem, as evidenced by the NP-hardness of optimal k-means clustering in the literature[3]. Researchers have devised heuristics and approximation algorithms to address this complexity, often employing Bayesian information criterion (BIC) for cluster count estimation[2, 4]. Our approach of searching through a range of clusters and selecting via BIC is a common practice despite it perhaps lacking finesse; see the recent review by Ezugwu et al.[5].
>
> In section 3.4.2, we have stated that we iterate n_components over 1 to m, not over 1 to 25. The implementation of PULSNAR provides a parameter, 'max_clusters', that one can set to any value to change the maximum number of clusters. The default value of this parameter is 25, and alternatives can be used to adapt to diverse datasets. Note that we apply clustering only to the positive set, and the number of labeled positives is usually very small compared to the unlabeled examples in any real-world data. Also, we use only important features scaled by their gain score in the clustering method. This makes the input data to the clustering algorithm relatively smaller. Additionally, our integration of multi-core machine learning tools from scikit-learn somewhat mitigates runtime concerns associated with determining the number of clusters. Despite an intensive search to determine the number of clusters, our algorithms ran faster than all the other compared ones; hence we did not focus on further optimizations.
>
> Since KM1, KM2, and TiCE could not be executed on large datasets, the following table shows the execution time for only DEDPUL and PULSNAR on 3 different synthetic datasets. DEDPUL was executed with CatBoost and 16 cores. PULSNAR was executed with XGBoost and 16 cores, and max_clusters set to 25. The execution time is based on a single execution of PULSNAR and DEDPUL to estimate $\alpha$. On large datasets, PULSNAR executes faster than DEDPUL despite determining the number of clusters and estimating $\alpha$ for each cluster.
>
>
> | Data size            | DEDPUL execution time                            | PULSNAR execution time                         |
> |----------------------|--------------------------------------------------|------------------------------------------------|
> | P: 5,000, U: 100,000 | 19m47.602s | 9m10.725s |
> | P: 5,000, U: 50,000  | 4m4.824s    | 1m37.448s   |
> | P: 5,000, U: 10,000  | 0m18.840s   | 1m6.786s   |
>
>
>
> [1] Song Xi Chen. Beta kernel estimators for density functions. Computational Statistics & Data Analysis, 31(2): 131–145, 1999.
>
> [2] Qinpei Zhao, Ville Hautamaki, and Pasi Fränti. Knee point detection in bic for detecting the number of clusters. In International conference on advanced concepts for intelligent vision systems, pp. 664–673. Springer, 2008.
>
> [3] Mahajan M, Nimbhorkar P, Varadarajan K. The planar k-means problem is NP-hard. Theor Comput Sci. 2012 Jul 13;442:13–21.
>
> [4] Fraley C, Raftery AE. How many clusters? Which clustering method? Answers via model-based cluster analysis. The computer journal. 1998 Jan 1;41(8):578-88.
>
> [5] Ezugwu AE, Ikotun AM, Oyelade OO, Abualigah L, Agushaka JO, Eke CI, Akinyelu AA. A comprehensive survey of clustering algorithms: State-of-the-art machine learning applications, taxonomy, challenges, and future research prospects. Engineering Applications of Artificial Intelligence. 2022 Apr 1;110:104743.

---

### Decision · Action_Editor_Ufwj · 2024-01-16

**Recommendation:** Reject

**Comment:**

In this paper, the authors propose two novel approaches to estimate positive proportion in PU learning under the SCAR and the SNAR settings respectively. It is an interesting topic which may contribute to related communities. During the discussions, some concerns about clarity of presentation, implementation details are successfully addressed.  However, the reviewers still have some common concerns that the experiments are not sufficient and convincing and the tightness of the upper bound presented in Equation 3 is not guaranteed.  Thus, the paper cannot be accepted in current version.

**Audience:**

The paper focuses on the class prior estimation problem when the SCAR assumption is not hold in PU learning, which is an interesting topic which may contribute to related communities. The main idea is to derive an upper bound as the class prior. The proposed PULSNAR algorithm adopts divide-and-conquer strategy which is interesting.

**Claims And Evidence:**

The tightness of the upper bound presented in Equation 3 is not guaranteed. The effectiveness of the proposed clustering approach to decompose the SNAR problem into SCAR problems is not convincingly demonstrated. The experiments are only performed on 2 real-world datasets, and the experiments are limited to small class prior value. More extensive experiments are needed.

**Resubmission Of Major Revision:**

The authors may consider submitting a major revision at a later time.